# MULTI-AGENT TRUST REGION LEARNING

## ABSTRACT

Trust-region methods are widely used in single-agent reinforcement learning. One advantage is that they guarantee a lower bound of monotonic payoff improvement for policy optimization at each iteration. Nonetheless, when applied in multi-agent settings, such guarantee is lost because an agent's payoff is also determined by other agents' adaptive behaviors. In fact, measuring agents' payoff improvements in multi-agent reinforcement learning (MARL) scenarios is still challenging. Although game-theoretical solution concepts such as Nash equilibrium can be applied, the algorithm (e.g., Nash-Q learning) suffers from poor scalability beyond two-player discrete games. To mitigate the above measurability and tractability issues, in this paper, we propose Multi-Agent Trust Region Learning (MATRL) method. MATRL augments the single-agent trust-region optimization process with the multi-agent solution concept of stable fixed point that is computed at the policy-space meta-game level. When multiple agents learn simultaneously, stable fixed points at the meta-game level can effectively measure agents' payoff improvements, and, importantly, a meta-game representation enjoys better scalability for multi-player games. We derive the lower bound of agents' payoff improvements for MATRL methods, and also prove the convergence of our method on the meta-game fixed points. We evaluate the MATRL method on both discrete and continuous multi-player general-sum games; results suggest that MATRL significantly outperforms strong MARL baselines on grid worlds, multi-agent MuJoCo, and Atari games.

## 1 INTRODUCTION

Multi-agent systems (MAS) (Shoham & Leyton-Brown, 2008) have received much attention from the reinforcement learning community. In real-world, automated driving (Cao et al., 2012), StarCraft II (Vinyals et al., 2019) and Dota 2 (Berner et al., 2019) are a few examples of the myriad of applications that can be modeled by MAS. Due to the complexity of multi-agent problems (Chatterjee et al., 2004), investigating if agents can learn to behave effectively during interactions with environments and other agents is essential (Fudenberg et al., 1998). This can be achieved naively through the *independent learner* (IL) (Tan, 1993), which ignores the other agents and optimizes the policy assuming a stable environment (Buşoniu et al., 2010; Hernandez-Leal et al., 2017). Due to their theoretical guarantee and good empirical performance in real-world applications, *trust region* methods (e.g., PPO (Schulman et al., 2015; 2017)) based ILs are popular (Vinyals et al., 2019; Berner et al., 2019). In single-agent learning, trust region methods can produce a monotonic payoff improvement guarantee (Kakade & Langford, 2002) via line search (Schulman et al., 2015).

In multi-agent scenarios, however, an agent's improvement is affected by other agent's adaptive behaviors (i.e., the multi-agent environment is *non-stationary* (Hernandez-Leal et al., 2017)). As a result, trust region learners can measure the policy improvements of the agents' current policies, but the improvements of the updated opponents' policies are unknown (shown in Fig. 1). Therefore, trust region based ILs act less well in MAS as in single-agent tasks. Moreover, the convergence to a *fixed point*, such as a *Nash equilibrium* (Nash et al., 1950; Bowling & Veloso, 2004; Mazumdar et al., 2020), is a common and widely accepted solution concept for multi-agent learning. Thus, although independent learners can best respond to other agents' current policies, they lose their convergence guarantee (Laurent et al., 2011).

One solution to address the convergence problem for independent learners is Empirical Game-Theoretic Analysis (EGTA) (Wellman, 2006), which approximates the best response to the policies generated by the independent learners (Lanctot et al., 2017; Muller et al., 2019). Although EGTA based methods (Lanctot et al., 2017; Omidshafiei et al., 2019; Balduzzi et al., 2019) establish

Figure 1: The relationship of discounted returns $\eta_i$ for an agent $i$ given the different joint policy pairs, where $\pi_i$ is the current policy, $\pi_i'$ is the simultaneously updated policy. Given $\pi_i$, the monotonic improvement against fixed opponent can be easily measured: $\eta_i(\pi_i', \pi_{-i}) \geq \eta_i(\pi_i, \pi_{-i})$. However, due to the simultaneous learning, the improvement of $\eta_i(\pi_i', \pi_{-i}')$ is unknown compared to $\eta_i(\pi_i, \pi_{-i})$.

convergence guarantees in several games classes, the computational cost is also large when empirically approximating and solving the increasing meta-game (Yang et al., 2019). Other multi-agent learning approaches collect or approximate additional information such as communication (Foerster et al., 2016) and centralized joint critics (Lowe et al., 2017; Foerster et al., 2017; Sunehag et al., 2018; Rashid et al., 2018). Nevertheless, these methods usually require centralized parameters or centralized communication assumptions. Thus, there is considerable interest in multi-agent learning to find an algorithm that, while having minimal requirements and computational cost as independent learners, also improves convergence performance at the same time.

This paper presents the *Multi-Agent Trust Region Learning* (MATRL) algorithm that augments the trust-region ILs with a meta-game analysis to improve the stability and efficiency of learning. In MATRL, a trust region trial step for an agents' payoff improvement is implemented by independent learners, which gives a predicted policy based on the current policy. Then, an empirical policy-space meta-game is constructed to compare the expected advantage of predicted policies with the current policies. By solving the meta-game, MATRL finds a restricted step by aggregating the current and predicted policies using meta-game Nash Equilibrium. Finally, MATRL takes the best responses based on the aggregated policies from last step for each agent to explore because the found TSR is not always strict stable. MATRL is, therefore, able to provide a weak stable solution compared with the naive independent learners. Based on trust region independent learners, MATRL does not need extra parameters, simulations, or modifications to the independent learner itself. We provide insights into the empirical meta-game in Section 3.2, showing that an approximated Nash equilibrium of the meta-game is a weak stable fixed point of the underlying game. Our experiments demonstrate that MATRL significantly outperforms deep independent learners (Schulman et al., 2017) with the same hyper-parameters, centralized VDN (Sunehag et al., 2018), QMIX (Rashid et al., 2018) methods in discrete action grid-worlds, centralized MADDPG (Lowe et al., 2017) in a continuous action multi-agent MuJoCo task (de Witt et al., 2020) and zero-sum multi-agent Atari (Terry & Black, 2020).

## 2 PRELIMINARY

A Stochastic Game (Shapley, 1953; Littman, 1994) can be defined as: $\mathcal{G} = \langle \mathcal{N}, \mathcal{S}, \{\mathcal{A}_i\}, \{\mathcal{R}_i\}, \mathcal{P}, p_0, \gamma \rangle$, where $\mathcal{N}$ is a set of agents, $n = |\mathcal{N}|$ is the number of agents and $\mathcal{S}$ denotes the state space. $\mathcal{A}_i$ is the action space for agent $i$. $\mathcal{A} = \mathcal{A}_1 \times \cdots \times \mathcal{A}_n = \mathcal{A}_i \times \mathcal{A}_{-i}$ is the joint action space, and for the simplicity we use $-i$ denotes the other agents except agent $i$. $\mathcal{R}_i = R_i(s, a_i, a_{-i})$ is the reward function for agent $i \in \mathcal{N}$. $\mathcal{P} : \mathcal{S} \times \mathcal{A} \times \mathcal{S} \to [0, 1]$ is the transition function. $p_0$ is the initial state distribution, $\gamma \in [0, 1]$ is a discount factor. Each agent $i \in \mathcal{N}$ has a stochastic policy $\pi_i(a_i|s) : \mathcal{S} \times \mathcal{A}_i \to [0, 1]$, and aims to maximize its long term discounted return:

$$\eta_i(\pi_i, \pi_{-i}) = \mathbb{E}_{s^0, a_i^0, a_{-i}^0 \cdots} \left[ \sum_{t=0}^{\infty} \gamma^t R_i(s^t, a_i^t, a_{-i}^t) \right], \tag{1}$$

where $s^0 \sim p_0$, $s^{t+1} \sim \mathcal{P}(s^{t+1}|s^t, a_i^t, a_{-i}^t)$, $a_i^t \sim \pi_i(a_i^t|\tau_i^t)$. We then can have the standard definitions of the state-action value function $Q_i^{\pi_i, \pi_{-i}}(s^t, a_i^t, a_{-i}^t) = \mathbb{E}_{s^{t+1}, a_i^{t+1}, a_{-i}^{t+1} \cdots}[\sum_{l=0}^{\infty} \gamma^l R_i(s^{t+l}, a_i^{t+l}, a_{-i}^{t+l})]$, the value function $V_i^{\pi_i, \pi_{-i}}(s^t) = \mathbb{E}_{a_i^t, a_{-i}^t, s^{t+1} \cdots}[\sum_{l=0}^{\infty} \gamma^l R_i(s^{t+l}, a_i^{t+l}, a_{-i}^{t+l})]$, and the advantage function $A_i^{\pi_i, \pi_{-i}}(s^t, a_i^t, a_{-i}^t) = Q_i^{\pi_i, \pi_{-i}}(s^t, a_i^t, a_{-i}^t) - V_i^{\pi_i, \pi_{-i}}(s^t)$ given the state and joint action.

## 3 MULTI-AGENT TRUST REGION POLICY OPTIMIZATION

A trust region algorithm aims to answer two questions: how to compute a trust region trial step and whether a trial step should be accepted. In multi-agent learning, a trust region trial step towards agents' payoff improvement can be easily implemented with independent learners, and we call the independent payoff improvement area as *Trust Payoff Region*(**TPR**). The remaining issue is

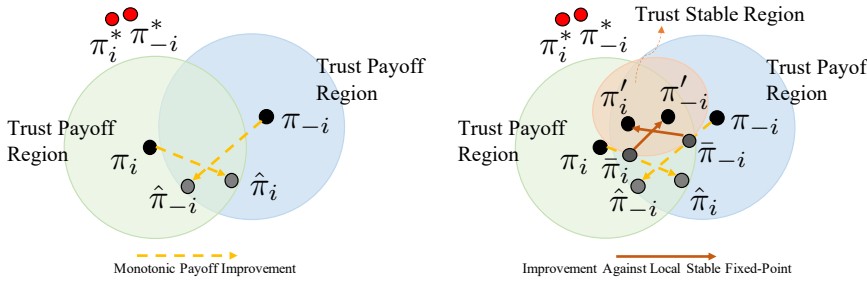

(a) Independent trust region learning.  (b) Multi-agent trust region learning.

Figure 2: Comparisons between independent trust region learner and multi-agent trust region learner. $\pi_i, \pi_{-i}$ are the current policies for two agents. $\hat{\pi}_i, \hat{\pi}_{-i}$ predicted policies within TPR, $(\pi_i^*, \pi_{-i}^*)$ forms Nash equilibrium, $\pi_i'$ and $\pi_{-i}'$ are the best responses to the weak stable fixed point $(\bar{\pi}_i, \bar{\pi}_{-i})$. **(a)**: independent trust region learning, an agent $i$ only considers itself's policy improvement against the fixed opponent policy $\pi_{-i}$. **(b)**: multi-agent trust region learning, agents' policy improvement should be explored in the joint policy space $(\pi_i, \pi_{-i})$ toward a stable region.

revolved by finding a restricted step leading to a stable point in the joint policy space, denoted as **Trust Stable Region**(**TSR**). In other words, multi-agent trust region learning (MATRL) decomposes the trust region learning into two parts: firstly, find a trust payoff region between *current policy* $\pi_i$ and *predicted policy* $\hat{\pi}_i$; then, with the help of the predicted policy, a precise method can, to some extent, approximate a *weak stable fixed point*. Instead of line searching in a single-agent payoff improvement, MATRL searches for the joint policy space to achieve a weak stable fixed point (see Fig. 2). Essentially, MATRL is a simple extension of the single-agent TRPO to MAS where independent learners learn to find a stable point between current policy and predicted policy. To solve the TSR, we assume the knowledge about other agents' policies during the training to find weak stable points via empirical meta-game analysis, while the execution can still be fully decentralized. We explain every step of MATRL in detail in the following sections.

## 3.1 INDEPENDENT TRUST PAYOFF IMPROVEMENT

Single-agent reinforcement learning algorithms can be straightforwardly applied to multi-agent learning, where we assume that all agents behave independently (Tan, 1993). In this section, we have chosen the policy-based reinforcement learning method as independent learners. In multi-agent games, the environment becomes a Markov decision process for agent $i$ when each of the other agents plays according to a fixed policy. We set agent $i$'s to make a monotonic improvement against the fixed opponent policies. Thus, at each iteration, the policy is updated by maximizing the utility function $\eta_i$ over a local neighborhood of the current joint policy $\pi_i, \pi_{-i}$: $\hat{\pi}_i = \arg\max_{\pi_i \in \Pi_i} \eta_i(\pi_i, \pi_{-i})$ based on the trajectories sampled by $\pi_i, \pi_{-i}$. We can adopt trust region policy optimization (e.g., PPO (Schulman et al., 2017)), which constrains step size in the policy update:

$$\hat{\pi}_i = \arg\max_{\pi_i \in \Pi_{\theta_i}} \eta_i(\pi_i, \pi_{-i}) \quad \text{s.t. } D(\pi_i, \hat{\pi}_i) \leq \delta_i, \tag{2}$$

where $D$ is a distance measurement and $\delta_i$ is a constraint. Independent trust region learners produce the monotonically improved policy $\hat{\pi}_i$ which guarantees $\eta_i(\hat{\pi}_i, \pi_{-i}) \geq \eta_i(\pi_i, \pi_{-i})$ and give a trust payoff bound by $\hat{\pi}_i$. Due to the simultaneous policy improvement without awareness of other agents , however, the lower bound of payoff improvement from single-agent Schulman et al. (2015) no longer holds for multi-agent payoff improvement. Following the similar proof procedures, we can obtain a precise lower bound for a multi-agent simultaneous trust payoff region in Theorem 1:

**Theorem 1** (Independent Trust Payoff Region). *Denote the expected advantage gain when $\pi_i, \pi_{-i} \rightarrow \hat{\pi}_i, \hat{\pi}_{-i}$ as:*

$$g_i^{\pi_i, \pi_{-i}}(\hat{\pi}_i, \hat{\pi}_{-i}) := \sum_s p^{\pi_i, \pi_{-i}}(s) \sum_{a_i} \hat{\pi}_i(a_i|s) \sum_{a_{-i}} \hat{\pi}_{-i}(a_{-i}|s) A_i^{\pi_i, \pi_{-i}}(s, a_i, a_{-i}). \tag{3}$$

*Let $\alpha_i = D_{\text{TV}}^{\max}(\pi_i, \hat{\pi}_i) = \max_s D_{\text{TV}}(\pi_i(\cdot|s)\|\hat{\pi}_i(\cdot|s))$ for agent $i$, where $D_{\text{TV}}$ is total variation divergence (Schulman et al., 2015). Then, the following lower bound can be derived for multi-agent independent trust region optimization:*

$$\eta_i(\hat{\pi}_i, \hat{\pi}_{-i}) - \eta_i(\pi_i, \pi_{-i}) \geq g_i^{\pi_i, \pi_{-i}}(\hat{\pi}_i, \hat{\pi}_{-i}) - \frac{4\gamma\epsilon_i}{(1-\gamma)^2}(\alpha_i + \alpha_{-i} - \alpha_i\alpha_{-i})^2, \tag{4}$$

*where $\epsilon_i = \max_{s, a_{-i}, a_i} |A_i^{\pi_i, \pi_{-i}}(s, a_i, a_{-i})|$.*

*Proof.* See Appendix B. □

Based on the independent trust payoff improvement, although the predicted policy $\hat{\pi}_i$ will guide us in determining the step size of the TPR, but the stability of $(\hat{\pi}_i, \hat{\pi}_{-i})$ is still unknown. As shown in Theorem 1, an agent's lower bound is roughly $O(4\alpha^2)$, which is four times larger than the single-agent lower bound trust region of $O(\alpha^2)$ (Kakade & Langford, 2002). Furthermore, $\epsilon_i = \max_{s, a_{-i}, a_i} \left| A_i^{\pi_i, \pi_{-i}}(s, a_i, a_{-i}) \right|$ depends on the other agents's action $a_{-i}$ that would be very large when agents have conflicting interests. Therefore, the most critical issue underlying the multi-agent trust region learning is to find a TSR after the TPR. In next section, we will illustrate how to search for a weak stable fixed point within the TPR, based on the policy-space meta-game analysis.

## 3.2 APPROXIMATING WEAK STABLE FIXED POINT

In multi-agent trust region learning, TSR is one of the essential parts. Since each iteration of MATRL requires solving the TPR and TSR sub-problems, finding the efficient solver for stable trust region sub-problems is very important. Instead of using the stable fixed points (Balduzzi et al., 2018) as TSR, we choose *weak stable fixed point* in Definition 1 which is much easier to be found. To maximize the objective defined in Eq. 1 we could ask that *reasonable* algorithms avoid all strict minimums (a.k.a unstable fixed points), which imposes only that agents are well-behaved regarding strict minimum even if their individual behaviors is not self-interested (Letcher, 2020), and we say a point is in TSR if it is weak stable fixed point.

**Definition 1** (Weak Stable Fixed Point in Restricted Underlaying Game). *Consider a **restricted underlying game**, where each agent's policy space is restricted to open sets $\bar{\Pi}_i = [\pi_i, \hat{\pi}_i] \subseteq \Pi_i$. Denote the simultaneous gradient of the restricted underlying game as $\boldsymbol{\xi} = (\nabla_{\pi_i} g_i, \nabla_{\pi_{-i}} g_{-i})$ and Hessian $H = \nabla \boldsymbol{\xi}$. We call $(\bar{\pi}_i, \bar{\pi}_{-i})$ a fixed point if $\boldsymbol{\xi}(\bar{\pi}_i, \bar{\pi}_{-i}) = \mathbf{0}$. We then say $(\bar{\pi}_i, \bar{\pi}_{-i})$ is a weak stable fixed point if $H(\bar{\pi}_i, \bar{\pi}_{-i}) \not\succ 0$[1], which avoids unstable fixed points (strict minimum). A trust stable region within weak stable fixed points is reasonable if it converges only to fixed points and avoids unstable fixed points almost surely.*

Given that we already have the TPR, which produces a predicted policy, with the knowledge about all the agents policies, it is natural to conduct an empirical game-theoretic analysis (Tuyls et al., 2018) to search for a weak stable fixed point in the area bounded by current policy pair predicted policy pair. We then define a meta-game that each agent $i$ has only two strategies $\pi_i, \hat{\pi}_i$ for each agent $i$:

$$\mathcal{M}(\pi_i, \hat{\pi}_i, \pi_{-i}, \hat{\pi}_{-i}) = \begin{pmatrix} g_i^{i,-i}, g_{-i}^{i,-i} & g_i^{i,-\hat{i}}, g_{-i}^{i,-\hat{i}} \\ g_i^{\hat{i},-i}, g_{-i}^{\hat{i},-i} & g_i^{\hat{i},-\hat{i}}, g_{-i}^{\hat{i},-\hat{i}} \end{pmatrix}, \tag{5}$$

where $g_i^{\hat{i},-\hat{i}} = g_i^{\pi_i, \pi_{-i}}(\hat{\pi}_i, \hat{\pi}_{-i})$ (as defined in Eq. 3) is an empirical payoff entry of the meta-game, and note $g_i^{i,-i} = 0$ as it has an expected advantage over itself. Compared with using the $\eta_i(\hat{\pi}_i, \hat{\pi}_{-i}) = \eta_i(\pi_i, \pi_{-i}) + g_i^{\hat{i},-\hat{i}}$ as the meta-game payoff, $g_i^{\hat{i},-\hat{i}}$ has lower variance and is easier to approximate because $\eta_i(\pi_i, \pi_{-i})$ is a constant baseline. However, the most of entries in $\mathcal{M}$ are unknown, it often requires lots of extra simulations to estimate the payoff entries (e.g., $g_i^{\hat{i},-\hat{i}}$) in EGTA. Instead, we reuse the trajectories in the TPR step to approximate the $g_i^{\hat{i},-\hat{i}}$ by ignoring small changes in state visitation density caused by the $\pi_i \rightarrow \hat{\pi}_i$ (Schulman et al., 2015).

Take two-agent case as an example, as we can see in Eq. 5, the meta-game $\mathcal{M}$ becomes a $2 \times 2$ matrix-form game, which is much smaller in size than the whole underlaying game. To this end, we can use the existing Nash solvers (e.g. CMA-ES (Hansen et al., 2003)) for matrix-form games to compute a mixed Nash equilibrium $\rho_i, \rho_{-i} = \text{NashSolver}(\mathcal{M})$ for the meta-game $\mathcal{M}$, where $\rho_i$ and $\rho_{-i} \in [0, 1]$, and the mixed Nash equilibrium of the meta-game is also an approximated equilibrium of the restricted underlying game (Tuyls et al., 2020). Then, the trust stable region policies $\bar{\pi}_i, \bar{\pi}_{-i}$ can be aggregated based on current policy $\pi_i$ and predicted policy $\hat{\pi}_i$ in TPR for each agent $i$. In the TPR step, we require ILs enjoy the monotonic improvement against the fixed opponent policies, in which the change from $\pi_i$ to $\hat{\pi}_i$ is usually constrained by a small step size. Therefore, it is reasonable to assume there is a continuous and monotonic change in the restricted policy space between $\pi_i$ and $\hat{\pi}_i$. In this case, with $\rho_i$ being agent $i$'s Nash Equilibrium policy in the meta-game, $\bar{\pi}_i$ can be derived via the linear mixture: $\bar{\pi}_i = \rho_i \pi_i + (1 - \rho_i)\hat{\pi}_i$, which delimit agent $i$'s trust stable region. Now we can prove that $(\bar{\pi}_i, \bar{\pi}_{-i})$ is a weak stable fixed point for the underlying game in Theorem 2.

---

[1] In this paper, we want to maximize the return, not minimizing the loss, so we need to avoid strict minimum.

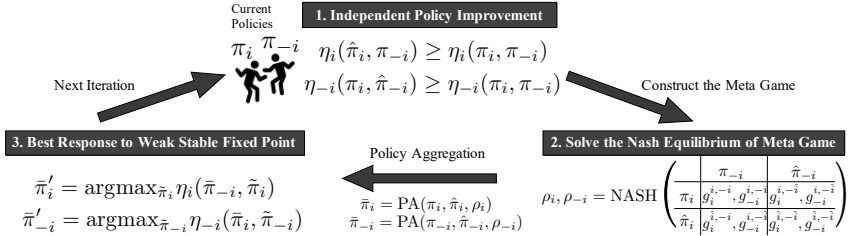

Figure 3: Overview of the multi-agent trust region learning phases in two-agent games. It can be easily extended to the $n$-agent case by solving the $n$-agent two-action matrix form meta-game.

---

**Algorithm 1** Multi-Agent Trust Region Learning

---

**Input:** Initializing policies $\pi_i$ for each $i$.
1: **for** $k \in \{0, 1, 2, \cdots\}$ **do**
2: Using current policies $\pi_i$, $\pi_{-i}$ to collect trajectories.
3: **For each** $i$: compute a trust payoff region policy $\hat{\pi}_i$ using Eq. 2.  ▷ Trust Payoff Region.
4: Solve meta-game $\mathcal{M}(\pi_i, \hat{\pi}_i, \pi_{-i}, \hat{\pi}_{-i})$ and obtain a meta-game Nash $\rho_i, \rho_{-i}$.
5: Compute weak stable fixed point $\bar{\pi}_i, \bar{\pi}_{-i}$.       ▷ Trust Stable Region.
6: **For each** $i$: compute best response $\pi_i'$ using Eq. 6.  ▷ Best Response to Fixed Point.
7:  $\pi_i \leftarrow \pi_i'$, $\pi_{-i} \leftarrow \pi_{-i}'$.
8: **end for**

---

**Theorem 2** (Existence of Weak Stable Fixed Point). *Consider the* restricted underlying game *where policy space is bounded in a linear continuous policy-space* $[\pi_i, \hat{\pi}_i]$, *where* $\hat{\pi}_i$ *is monotonically improved based on* $\pi_i$ *within TPR. If* $(\rho_i, \rho_{-i})$ *is a Nash equilibrium of the meta-game* $\mathcal{M}$, *then, the linear mixture joint policy* $(\bar{\pi}_i, \bar{\pi}_{-i})$ *is a weak stable fixed point for the restricted underlying game.*

*Proof.* see Appendix C.                     □

According to Theorem 2, $(\bar{\pi}_i, \bar{\pi}_{-i})$ is a weak stable fixed point of the restricted underlying game. Although the weak stable fixed point is relatively weak compared to the stable fixed points (Balduzzi et al., 2018), as we have stated, a weak stable fixed point is a reasonable (not strong as rational) requirement for an algorithm to avoid the minimum. Furthermore, the weak stable fixed points can suit for general game settings. As shown in Appendix C, in cooperative, competitive, and general-sum games, the fixed-point found by the meta-game analysis can be either stable or saddle points. Similarly, a local Nash equilibrium can be a stable or saddle in different differential games (Mazumdar et al., 2020). Therefore, the goodness of stable concepts would depend on specific settings. If we make some additional game class assumptions, we can easily obtain a stronger fixed-point types. Nevertheless, it comes with a cost, requiring additional computation or assumptions which may break the most general settings. Besides, when the meta-game has multiple Nash equilibria, an equilibrium is randomly selected in our work. Some equilibria can produce a more stable fixed point, however, we leave the equilibrium selection problem for future work.

**Extra Cost for Approximating and Solving meta-game**. There are two major-cost sources in common meta-game analysis: approximating and solving the meta-game (Muller et al., 2019). In our case, the meta-game is restricted to a local two-action game, where two actions $\pi_i$ and $\hat{\pi}_i$ are close to each other. This proximity property reduces the meta-game approximation cost (without extra sampling) by reusing the collected trajectories in the TPR step (Tuyls et al., 2020). The next crucial problem is how to solve the $n$-agent two-action meta-game, which consists of the $2^n$ entries of each of the $n$ payoff matrices. This is much simpler than solving the whole underlying game, which increases exponentially with state size, action size, agent number, and time horizons. As the general-sum matrix-form game has no fully polynomial-time approximation for computing Nash equilibria (Chen et al., 2006), it usually costs a lot to solve the game (Daskalakis et al., 2009). If we only require an approximated Nash equilibrium, when $n$ is small, for example, $n \leq 10$, it is affordable to find a meta-game Nash equilibrium in a sub-exponential complexity (Lipton et al., 2003). However, this problem still exists when $n$ is large. In this case, we could try mean-field approximation (Yang et al., 2018) or utilizing special payoff structure assumptions (e.g., graphical game (Littman et al., 2002; Daskalakis et al., 2009), which is polynomial-time computable.) in the meta-game to reduce the computation complexity.

### 3.3 IMPROVEMENT AGAINST WEAK STABLE FIXED POINT

Although the weak stable fixed point, $(\bar{\pi}_i, \bar{\pi}_{-i})$, binds the policy update to another fixed point, there are still undesired saddle points according to Theorem 1. It is difficult to generalize for the other parts of the policy-space not reached by these saddle points, especially in the anti-coordination games (Lanctot et al., 2017). Similar to the extra-gradient method (Mertikopoulos et al., 2018), to escape the saddle points we apply the best response against the weak stable fixed point:

$$\pi'_i = \arg\max_{\pi_i} \eta_i\left(\pi_i, \bar{\pi}_{-i}\right). \tag{6}$$

To perform the best response, we need another round to collect the experiences and do a gradient step in Eq. 6. However, in practice, since we already have the trajectories in the TPR step and so the best response to the weak stable fixed point can be easily estimated through importance sampling. Alternatively, through defining $c_i \overset{\text{def}}{=} \min\left(1 + \bar{c}, \max(1 - \bar{c}, \frac{\pi_i(a_i|s)}{\bar{\pi}_i(a_i|s)})\right)$ as truncated importance sampling weights, we can re-write the best response update to Eq. 6 into an equivalent form to the following one in terms of expectations: $\pi'_i = \arg\max_{\pi_i} \mathbb{E}_{a_{-i} \sim \bar{\pi}_{-i}}[c_{-i}\eta_i\left(\pi_i, \pi_{-i}\right)]$.

**Connections to Existing Methods**. MATRL generalizes many existing methods with the best response. In extreme cases where the meta-game Nash is $(\rho_i, \rho_{-i}) = (1, 1)$, which means the Nash aggregated policies always keep the current policies, MATRL degenerates to independent learners. Here, we always best response to the other agents' current policy $\pi_i$ and $\pi'_i = \arg\max_{\pi_i} \eta_i(\pi_i, \pi_{-i})$ following the Eq. 6. The policy prediction (Zhang & Lesser, 2010; Foerster et al., 2018; Letcher et al., 2018), extra-gradient (Antipin, 2003) and exploitability descent (Tang et al., 2018; Lockhart et al., 2019) methods are also special instances of MATRL when meta-game Nash is $(\rho_i, \rho_{-i}) = (0, 0)$. This means the best response to the most aggressive predicted policy $\hat{\pi}_{-i}$ and $\pi'_i = \arg\max_{\pi_i} \eta_i(\pi_i, \hat{\pi}_{-i})$.

**Global Convergence**. MATRL is a gradient-based algorithm with the best response to policies within TSR, which is essentially a variant of LookAhead methods(e.g., LOLA (Foerster et al., 2018), SOS (Letcher et al., 2018) and IGA-PP (Zhang & Lesser, 2010). More specifically, MATRL enhances the classic LookAhead method with variable step size scaling (Bowling & Veloso, 2002) or two time-scale update rule (Heusel et al., 2017) at each TSR step, which is controlled by the restricted meta-game analysis. It has been proven that LookAhead method can locally converge and avoid strict saddles in all differentiable games (Letcher et al., 2018), and enjoys the better convergence with variable step size scaling (Song et al., 2019). The convergence analysis of gradient-based algorithms is usually based on fixed-point iterations and dynamical systems. And please note, here, to investigate the convergence, the fixed-point iterations are conducted on the whole learning process. While the meta-game analysis step in MATRL borrows the concepts of different fixed-points to show the meta-game analysis is reasonable to avoid unstable fixed-points. Unlike LOLA, which uses first-order Taylor expansion to estimate the best response to a predicted policy, we elaborately design the look-ahead step within the TSR and do the best response gradient steps to TSR. We also show that MATRL empirically outperforms the typical LookAhead method independent learner the policy prediction (IL-PP) in the experiments.

In summary, independent trust region learners' learning in MTARL will be constrained by a weak stable fixed point. By analyzing the relatively simpler meta-game, we can easily approximate this weak stable fixed point without extra rollouts or simulation. Although MATRL's training is centralized, its execution is fully decentralized. It also does not require any extra centralized parameters or higher-order gradient computation. Fig. 3 shows the overview of MATRL. We also give the pseudocode of MATRL in Algo. 1, which is compatible with any policy-based independent learner.

## 4 RELATED WORK

The study of gradient-based methods in multi-agent learning is quite extensive (Mazumdar et al., 2020; Buşoniu et al., 2010). Some works on learning in games have mostly focused on adjusting the step size, which attempts to use a multiple-timescales learning scheme (Leslie & Collins, 2005; Leslie et al., 2003; Bowling & Veloso, 2002) to achieve convergence. Balduzzi et al. (2018); Mazumdar et al. (2019); Letcher et al. (2018) tried to utilize the second-order methods to shape the step size. However, the computational cost for second-order methods is very limiting in many cases. Alternatively, MATRL approximates the second-order fixed-point information via a small meta-game with less cost comparing to real Hessian computation. An alternative augments the gradient-based algorithms with the best response to predicted polices (Antipin, 2003; Zhang & Lesser, 2010; Lin et al., 2020; Foerster et al., 2018; Tang et al., 2018; Lockhart et al., 2019),

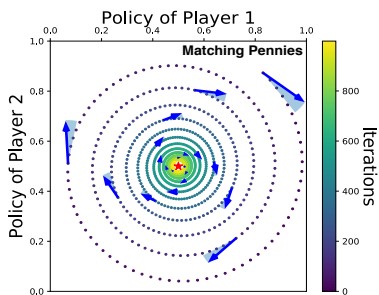

Figure 4: Learning dynamics of MATRL in matching pennies game. The blue arrow is gradient direction and the pale blue area is TSR.

Table 1: Convergence rate and average convergence step in 1000 random $2 \times 2$ matrix games. MATRL shows slightly better convergence rate and speed compared to IGA-PP.

| | CONVERGENCE RATE / AVERAGE CONVERGENCE STEP | | |
|---|---|---|---|
| ALGORITHM | COORDINATION | ANTI-COORD. | CYCLIC |
| IGA | **0.99** / 140.67 | 0.975 / 88.95 | 0.78 / 452.92 |
| IGA-PP | **0.99** / 138.56 | 0.975 / 83.11 | 0.809 / 432.98 |
| MATRL | **0.99** / **86.54** | **0.9825** / **75.52** | **0.846** / **369.40** |

which target the challenge of instability caused by agents' change policies. Instead of taking the best response to the approximated opponent's policy, MATRL exploits the ideas from both streams and and introduces the improvement over the weak stable fixed point.

The research also focuses on the EGTA (Tuyls et al., 2018; Jordan & Wellman, 2009; Tuyls et al., 2020), which creates a policy-space meta-game for modeling the multi-agent interactions. Using various evaluation metrics, it then updates and extends the policies based on the analysis of the meta policies (Lanctot et al., 2017; Muller et al., 2019; Omidshafiei et al., 2019; Balduzzi et al., 2019; Yang et al., 2019). Although these methods are broad with respect to multi-agent tasks, they require extensive computing resources to estimate the empirical meta-game and solve it with its increasing size (Omidshafiei et al., 2019; Yang et al., 2019). In our method, we adopt the idea of a policy-space meta-game to approximate the fixed point. Unlike previous works, we only maintain the current policies and predicted policies to construct the meta-game, which is computationally achievable in most cases. The payoff entry in MATRL's meta-game is the expected advantage, which has a lower estimation variance compared to the commonly used empirically-estimated return in EGTAs. Regardless, we can reuse the trajectories in the TPR step to estimate the payoffs without incurring additional sampling costs.

Recently, due to the use of neural networks as a function approximation for policies and values, there have emerged many works on deep reinforcement learning (DRL) (Mnih et al., 2013; Lillicrap et al., 2015). Trust region policy optimization (Kakade & Langford, 2002; Schulman et al., 2015; 2017) is one of the most successful DRL methods in the single-agent setting, which puts constraints on the step size of policy updates, preserving any improvements monotonically. Based the monotonic improvement in single-agent trust region policy optimization (TRPO) (Schulman et al., 2015), MATRL extends the improvement guarantee to the multi-agent level, towards a weak stable fixed point. Some works directly apply fully decentralized single-agent DRL methods (Tan, 1993), which can be unstable during when learning due to the non-stationary issue. Whereas (Foerster et al., 2016; Sukhbaatar et al., 2016; Peng et al., 2017) added an extra communication channel during the training and execution in a centralized way to avoid this non-stationarity issue. Sunehag et al. (2018); Rashid et al. (2018); Foerster et al. (2017); Lowe et al. (2017) further exploit the setting of centralized learning decentralized execution (CTDE). These methods provide solutions for training agents in complex multi-agent environments, and the experimental results show the effectiveness compared with independent learners. Similar to the CTDE setting, the MATRL also enjoy fully decentralized execution. Although MATRL still needs knowledge about the other agents' policies during the training, it only requires a centralized mechanism to adjust the step size rather than the additional centralized critic or communication.

## 5 EXPERIMENTS

We design the experiments to answer the following questions: 1). Can the MATRL method empirically contribute to the convergence in the general game settings, including cooperative/competitive and continuous/discrete games? 2). How is the performance of MATRL compared to the ILs with the same hyper-parameters and other strong MARL baselines in discrete and continuous games with various agent number? 3). Do the meta-game and best response to the weak stable fixed point bring benefits? We first evaluate the convergence performance of MATRL in matrix form games to answer the first question and validate the effectiveness of convergence. For Question 2, we show that MATRL largely outperforms ILs (PPO (Schulman et al., 2017)) and other centralized baselines (QMIX (Rashid et al., 2018), QTRAN (Son et al., 2019) and VDN (Sunehag et al., 2018)) for discrete grid world games

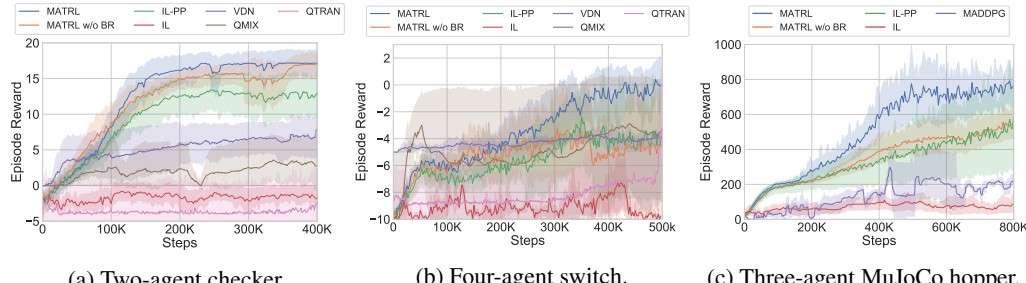

(a) Two-agent checker.    (b) Four-agent switch.    (c) Three-agent MuJoCo hopper.

Figure 5: Learning curves in discrete and continuous tasks. The solid lines are average episode returns with 10 random seeds for each model, and the light color areas are the error bar.

that have coordination problems. It also outperforms MADDPG (Lowe et al., 2017) for continuous multi-agent MuJoCo games. Besides, we test the algorithms with 2-agent pong Atari game to investigate if MATRL can mitigate unstable cyclic behaviors (Balduzzi et al., 2019) in zero-sum games. In these tasks, MATRL uses the same PPO configurations as ILs to examine the effectiveness of the trust region gradient-update mechanism, and we use official implementations for the other baselines. The step-by-step PPO based MATRL algorithm is given in Algo. 2. Finally, ablation studies are conducted by: 1. removing the best response, called the MATRL w/o BR; 2. skipping the trust-stable region estimation, named IL-PP, which has similar procedures as LOLA Foerster et al. (2018); Zhang & Lesser (2010), which approximated the best response to the predicted policies via Taylor expansion, but IL-PP takes the best response gradient steps to the predicted policies. These configurations provide insights about how much does the trust stable region and the best response contribute to the MATRL's performance if any. We also provide more environment details and extra experiment results, including 4-agent Ant (multi-agent MuJoCo), in the Appendix D and E with detailed experiment settings and hyper-parameters used for the algorithms. The code and experiment scripts are also anonymously available at https://github.com/matrl-project/matrl.

**Matrix Game and Random $2 \times 2$ Matrix Games**. To illustrate the effectiveness of MATRL, we conducted an experiment on well known zero-sum matching pennies (MP) (Bruns, 2015) game and devise the $2 \times 2$ random matrix games. Using IGA (Singh et al., 2000) as ILs of MATRL, the learning dynamics of MATRL on MP are shown in Fig. 4, where the blue arrow is trust payoff direction and the pale blue area is TSR.. The MATRL reaches the Nash Equilibrium (central red star point) by updating the policies with the constraints from the trust stable region (the pale blue area). It would be trapped to a cyclic loop if following the original trust pay off direction (the dark blue arrow). To adequately examine the MATRL on border matrix games, we randomly generate three thousand $2 \times 2$ games for three types: coordination, anti-coordination, and cyclic (Pangallo et al., 2017). More details about the game generation are provided in Section D. We choose the IGA and IGA-PP (Zhang & Lesser, 2010) as baselines, and the results in Table 1 show that MATRL has a higher convergence rate and needs fewer steps for convergence in all types of games.

**Grid World Checker and Switch**. We evaluated MATRL in two grid world games from MA-Gym (Koul, 2019), two-agent checker, and four-agent switch, which are similar to games in Sunehag et al. (2018), but with more agents to examine if the MATRL can handle the games that have more than two agents. In the checker game, two agents cooperate in collecting fruits on the map; the sensitive agent gets 5 for apple and −5 for lemon, while the other one gets 1 and −1 respectively. So the optimal solution is to let the sensitive agent get the apple and the less sensitive one get the lemon. In the four-agent switch game, two rooms are connected with a corridor, each room has two agents, and the four agents try to go through one corridor to the target in the opposite room. Only one agent can pass the corridor at one time, and agents get −0.1 for each step and 5 for reaching targets, so they need to cooperate to get optimal scores. In both games, The agents can move in four directions and only partially observe their position. Although our formulation uses a fully observable setting, in this game, the methods adopt to the partially observable by pretending the observation is a state. We compare the MATRL with the PPO based IL and two off-policy centralized training and decentralized execution baselines: VDN (Sunehag et al., 2018), QTRAN (Son et al., 2019) and QMIX (Rashid et al., 2018). Results are given in Fig. 5a and 5b, where MATRL has a stable improvement and outperforms other baselines. In two-player checker, using the best response, our method can achieve a total reward of 18, while the independent learners' rewards stay at −2. Besides, although PPO-based MATRL uses on-policy learning, it achieved better final results in fewer time steps compared to the off-policy baselines. As for the four-player switch, as shown in Fig. 5b, MATRL can continuously improve the total rewards to 6.5, which is the closest to the optimal score for this game when compared with

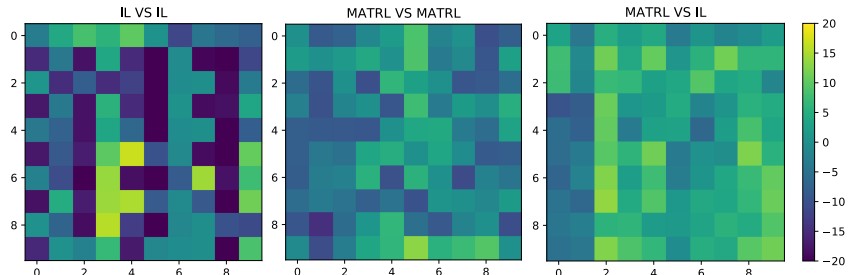

Figure 6: MATRL/IL versus MATRL/IL in the two-agent pong game. For each setting, the grids are pair-wise performance (average scores) by pitting their ten checkpoints against one another, yellow means higher score.

other baselines. The result in the four-agent switch also demonstrates the effectiveness of MATRL in guaranteeing the stable policy improvement for the games that have more than two agents.

**Multi-Agent MuJoCo Hopper**. We also examined MATRL in a multi-agent continuous control task with a three-agent hopper from (de Witt et al., 2020). Here, three agents cooperatively control each part of a hopper to move forward. The agents are rewarded with distance and the number of steps they make before falling. Fig. 5c shows that MATRL significantly outperforms IL, MADDPG, and also the benchmarks in de Witt et al. (2020) within the same amount of time.

**Multi-Agent Pong Atari Game**. In the 2-agent pong game experiments, we used raw pixel as observation and train the MATRL and IL agents independently. Following training, we compare these models' pair-wise performance by pitting their ten checkpoints against one another and recording the average scores. We report the results in Fig. 6, which shows MATRL outperforms IL in MATRL vs. IL setting in most of the policy pairs. Besides, from the MATRL vs. MATRL and IL vs. IL settings' results, we can see MATRL has a more transitive learning process than IL, which means MATRL can mitigate the common cyclic behaviors in zero-sum games.

**Effect and Cost of Trust Stable Region and Best Response to Fixed Point**. This section analyzes the effect of the TSR from meta-game Nash and the best response against the weak stable fixed point. The ablation settings are obtained by removing the trust stable region (IL-PP) and the best response (MATRL w/o BR). In Fig. 5, we can observe that in all the tasks, without the best response to the fixed point, the learning curves of MATRL o/w BR have higher variance and the lowest final scores. This establishes the importance of the best response to stabilize and improve agents' performance, and empirically shows that the MATRL has better convergence ability than other baselines. Also, without the

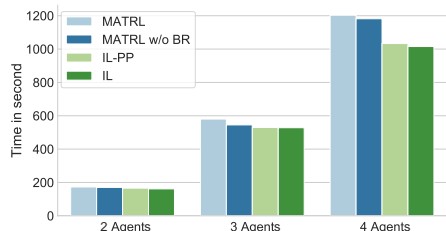

Figure 7: Running time of 20,000 environment steps (including 50 gradient steps) for the algorithms in 2-4 agents games.

TSR to select a fixed point, the MATRL recovers to independent learners with the policy prediction (IL-PP) (Zhang & Lesser, 2010; Foerster et al., 2018). Similarly, the curves of IL-PP have lower final scores, and the convergence speed is not as good as the MATRL, which suggests that the TSR provides benefits. The MATRL w/o BR has lower variance compared to the IL-PP, which reveals the trust stable region can stabilize the learning via weak stable fixed point constraints. Finally, when comparing to IL and IL-PP, as shown in Fig. 7, in 2-4 agents games with 20,000 environment steps and 50 gradient steps, the training time of MATRL is empirically about 1.1-1.2 times slower. We think this extra computational cost from the TSR and the best response is acceptable given the performance improvement brought by these operations.

## 6 CONCLUSION

We proposed and analyzed the trust region method for multi-agent learning problems, which considers the trust payoff region and the trust stable region to meet the multi-agent learning objectives. In practice, based on independent trust payoff learners, we provide a convenient way to approximate a further restricted step size within TSR via the meta-game. This ensures that the MATRL is generalized, flexible, and easily implemented to deal with multi-agent learning problems in general. Our experimental results justify the fact that MATRL method significantly outperforms independent learners using the same configurations, and other strong MARL baselines on both continuous and discrete games with various agent numbers.

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

## A  MATRL Algorithm Based on PPO

---

**Algorithm 2** Multi-Agent Trust Region Learning Algorithm (PPO Based, Two-Agent Example).

---

**Input:** The initial policy parameters $\theta_1, \theta_2$, initial value function parameters $\phi_1, \phi_2$ and $\epsilon$.

1: **for** $k \in \{0, 1, 2, \cdots\}$ **do**
2:     Using $\pi_1(\theta_1), \pi_2(\theta_2)$ to collect trajectories $\boldsymbol{\tau}_1, \boldsymbol{\tau}_2$.
3:     Compute GAE reward $\hat{R}_i$ for each $i$.
4:     Compute estimated advantages $\hat{A}_1, \hat{A}_2$ based on the current value functions $V_{\phi_1}, V_{\phi_2}$.
5:     **for** $i \in \{1, 2\}$ **do**
6:         Compute a trust payoff region policy $\hat{\pi}_i$ using Eq. 2.
7:         Update    the    policy    by    maximizing    the    PPO-Clip    objective:
$\hat{\theta}_i = \arg\max_{\theta_i} \frac{1}{|\boldsymbol{\tau}_i|T} \sum_{\tau \in \boldsymbol{\tau}_i} \sum_{t=0}^{T} \min\left( \frac{\pi_i(a_t|s_t;\theta)}{\pi_i(a_{1,t}|s_t;\theta_i)} A_i^{\pi_1,\pi_2}(s_t, a_{1,t}, a_{2,t}), \quad g\left(\epsilon, A_i^{\pi_1,\pi_2}(s_t, a_{1,t}, a_{2,t})\right)\right)$,
        where $g$ is a clipping function.
8:         Fit value function by regression on mean-squared error:

$$\phi_i' = \arg\min_{\phi_i} \frac{1}{|\boldsymbol{\tau}_i|T} \sum_{\tau \in \boldsymbol{\tau}_i} \sum_{t=0}^{T} \left( V_\phi(s_t) - \hat{R}_{i,t} \right)^2$$

9:     **end for**
10:     Construct the meta-game $\mathcal{M}(\pi_1(\theta_1), \hat{\pi}_1(\hat{\theta}_1), \pi_2(\theta_2), \hat{\pi}_2(\hat{\theta}_2))$.
11:     Solve $\mathcal{M}$ and obtain meta Nash $\rho_1, \rho_2$.
12:     Compute aggregated weak stable fixed point $(\bar{\pi}_1, \bar{\pi}_2)$.
13:     **for** $i \in \{1, 2\}$ **do**
14:         Compute $\pi_i^{(\prime)}$ which best responses to $\bar{\pi}_{-i}$ using Eq. 6.
15:         Estimate the best response by importance sampling:

$$\theta_i' = \frac{\hat{\theta}_i}{|\boldsymbol{\tau}_i|T} \sum_{\tau \in \boldsymbol{\tau}_i} \sum_{t=0}^{T} g\left(\epsilon, \pi_i / \bar{\pi}_{-i}\right)$$

16:     **end for**
17:     $\theta_1 \leftarrow \theta_1', \theta_2 \leftarrow \theta_2'$ .
18: **end for**
**Output:** $\pi_1(\theta_1), \pi_2(\theta_2)$.

---

## B  Independent Trust Payoff Region

We use the total variation divergence, which is defined by $D_{\text{TV}}(p\|q) = \frac{1}{2}\sum_j |p_j - q_j|$ for discrete probability distributions $p, q$ (Schulman et al., 2015). $D_{\text{TV}}^{\max}(\pi, \tilde{\pi})$ is defined as:

$$D_{\text{TV}}^{\max}(\pi, \tilde{\pi}) = \max_s D_{\text{TV}}(\pi(\cdot|s)\|\tilde{\pi}(\cdot|s)). \tag{7}$$

Based on this, we can define $\alpha$-coupled policy as:

**Definition 2** ($\alpha$-Coupled Policy (Schulman et al., 2015))**.** $(\pi, \pi')$ *is an $\alpha$-coupled policy pair if it defines a joint distribution $(a, a')|s$, such that $P(a \neq a'|s) \leq \alpha$ for all $s$. $\pi$ and $\pi'$ will denote the marginal distributions of $a$ and $a'$, respectively.*

When the joint policy pair $\pi_i, \pi_{-i}$ changes to $\pi_i', \pi_{-i}'$ and coupled with $\alpha_i$ and $\alpha_{-i}$ correspondingly:

$$\eta_i(\pi_i', \pi_{-i}') - \eta_i(\pi_i, \pi_{-i}) \geq A_i^{\pi_i, \pi_{-i}}(\pi_i', \pi_{-i}') - \frac{4\gamma\epsilon}{(1-\gamma)^2}(\alpha_i + \alpha_{-i} - \alpha_i\alpha_{-i})^2, \tag{8}$$

where

$$\epsilon = \max_{s, a_i, a_{-i}} \left| A_i^{\pi_i, \pi_{-i}}(s, a_i, a_{-i}) \right|.$$

The proofs are as following:

**Lemma 1.** *Given that $(\pi_i, \pi'_i)$ and $(\pi_{-i}, \pi'_{-i})$ are both $\alpha$-coupled policies bounded by $\alpha_i$ and $\alpha_{-i}$ respectively, for all $s$,*

$$\left| A_i^{\pi_i, \pi_{-i}}(s) \right| \le 2(\alpha_i + \alpha_{-i} - \alpha_i \alpha_{-i}) \max_{s, a_{-i}, a_{-i}} \left| A_i^{\pi_i, \pi_{-i}}(s, a_i, a_{-i}) \right| \tag{9}$$

*Proof.*

$$A_i^{\pi_i, \pi_{-i}}(s) = \mathbb{E}_{a'_i, a'_{-i} \sim \pi'_i, \pi'_{-i}} \left[ A_i^{\pi_i, \pi_{-i}}(s, a'_i, a'_{-i}) \right] \tag{10}$$

$$= \mathbb{E}_{(a_i, a'_i) \sim (\pi_i, \pi'_i), (a_{-i}, a'_{-i}) \sim (\pi_{-i}, \pi'_{-i})} \left[ A_i^{\pi_i, \pi_{-i}}(s, a'_i, a'_{-i}) - A_i^{\pi_i, \pi_{-i}}(s, a_i, a_{-i}) \right] \tag{11}$$

$$= P(a_i \ne a'_i \vee a_{-i} \ne a'_{-i} | s) \mathbb{E}_{(a_i, a'_i) \sim (\pi_i, \pi'_i), (a_{-i}, a'_{-i}) \sim (\pi_{-i}, \pi'_{-i})} [A_i^{\pi_i, \pi_{-i}}(s, a'_i, a'_{-i}) \tag{12}$$

$$- A_i^{\pi_i, \pi_{-i}}(s, a_i, a_{-i})] \tag{13}$$

$$\le (\alpha_i + \alpha_{-i} - \alpha_i \alpha_{-i}) \cdot 2 \max_{s, a_{-i}, a_{-i}} \left| A_i^{\pi_i, \pi_{-i}}(s, a_i, a_{-i}) \right|, \tag{14}$$

where $P(a_i \ne a'_i \vee a_{-i} \ne a'_{-i} | s) = 1 - (1 - \alpha_i)(1 - \alpha_{-i}) = \alpha_i + \alpha_{-i} - \alpha_i \alpha_{-i}$.

$\square$

**Lemma 2.** *Let $(\pi_i, \pi'_i)$ and $(\pi_{-i}, \pi'_{-i})$ are $\alpha$-coupled policy pairs. Then,*

$$\left| \mathbb{E}_{s_t \sim \pi'_i, \pi'_{-i}} \left[ A_i^{\pi_i, \pi_{-i}}(s) \right] - \mathbb{E}_{s_t \sim \pi_i, \pi_{-i}} \left[ A_i^{\pi_i, \pi_{-i}}(s) \right] \right|$$
$$\le 4(\alpha_i + \alpha_{-i} - \alpha_i \alpha_{-i})(1 - (1 - \alpha_i)^t (1 - \alpha_{-i})^t) \max_{s, a_{-i}, a_{-i}} \left| A_i^{\pi_i, \pi_{-i}}(s, a_i, a_{-i}) \right| \tag{15}$$

*Proof.* The preceding Lemma bounds the difference in expected advantage at each time step $t$. When $t' = 0$ indicates that $\pi_i, \pi_{-i}$ and $\pi'_i, \pi'_{-i}$ both agreed on all time steps less than $t$. By the definition of $\alpha_i, \alpha_{-i}$, $P(\pi_i, \pi_{-i} := \pi'_i, \pi'_{-i} | t = i) \ge (1 - \alpha_i)(1 - \alpha_{-i})$, so $P(t' = 0) \ge (1 - \alpha_i)^t (1 - \alpha_{-i})^t$ and $P(t' > 0) \le 1 - (1 - \alpha_i)^t (1 - \alpha_{-i})^t$. We can sum over time to bind the difference between $\eta_i(\pi'_i, \pi'_{-i})$ and $\eta_i(\pi_i, \pi_{-i})$.

$$\left| \eta_i(\pi'_i, \pi'_{-i}) - L_i^{\pi_i, \pi_{-i}}(\pi'_i, \pi'_{-i}) \right| = \sum_{t=0}^{\infty} \gamma^t \left| \mathbb{E}_{s_t \sim \pi'_i, \pi'_{-i}} \left[ A_i^{\pi_i, \pi_{-i}}(s) \right] - \mathbb{E}_{s_t \sim \pi_i, \pi_{-i}} \left[ A_i^{\pi_i, \pi_{-i}}(s) \right] \right| \tag{16}$$

$$\le \sum_{t=0}^{\infty} \gamma^t \cdot 4\epsilon(\alpha_i + \alpha_{-i} - \alpha_i \alpha_{-i})(1 - (1 - \alpha_i)^t (1 - \alpha_{-i})^t) \tag{17}$$

$$= 4\epsilon(\alpha_i + \alpha_{-i} - \alpha_i \alpha_{-i}) \left( \frac{1}{1 - \gamma} - \frac{1}{1 - \gamma(1 - \alpha_i)(1 - \alpha_{-i})} \right) \tag{18}$$

$$= \frac{4\epsilon(\alpha_i + \alpha_{-i} - \alpha_i \alpha_{-i})^2}{(1 - \gamma)(1 - \gamma(1 - \alpha_i)(1 - \alpha_{-i}))} \tag{19}$$

$$\le \frac{4\epsilon(\alpha_i + \alpha_{-i} - \alpha_i \alpha_{-i})^2}{(1 - \gamma)^2}, \tag{20}$$

where $\epsilon = \max_{s, a_i, a_{-i}} \left| A_i^{\pi_i, \pi_{-i}}(s, a_i, a_{-i}) \right|$. $\square$

Note that

$$L_i^{\pi_i, \pi_{-i}}(\pi'_i, \pi'_{-i}) = \eta_i(\pi_i, \pi_{-i}) + \sum_s \rho^{\pi_i, \pi_{-i}}(s) \sum_{a_i} \pi'_i(a_i | s) \sum_{a_{-i}} \pi'_{-i}(a_{-i} | s) A_i^{\pi_i, \pi_{-i}}(s, a_i, a_{-i}). \tag{21}$$

Then, we can have

$$\eta_i(\pi'_i, \pi'_{-i}) - \eta_i(\pi_i, \pi_{-i}) \ge A_i^{\pi_i, \pi_{-i}}(\pi'_i, \pi'_{-i}) - \frac{4\gamma\epsilon}{(1 - \gamma)^2} (\alpha_i + \alpha_{-i} - \alpha_i \alpha_{-i})^2. \tag{22}$$

## C  PROOF OF THEOREM 2

At each iteration, denote $\nabla_i g_i = \nabla_{\pi_i} g_i^{\pi_i, \pi_{-i}}$ and $\nabla_{i,-i} g_i = \nabla_{\pi_i} \nabla_{\pi_{-i}} g_i^{\pi_i, \pi_{-i}}$ for each $i$. Consider the simultaneous gradient $\boldsymbol{\xi}$ of the expected advantage gains and the corresponding Hessian $H$:

$$\boldsymbol{\xi}(\pi_i, \pi_{-i}) = (\nabla_i g_i, \nabla_{-i} g_{-i}), \tag{23}$$

$$H = \nabla \xi = \begin{pmatrix} \nabla_{i,i} g_i & \nabla_{i,-i} g_i \\ \nabla_{-i,i} g_{-i} & \nabla_{-i,-i} g_{-i} \end{pmatrix}. \tag{24}$$

For a restricted underlying game, where policy space is bounded: $\pi_i \in [\pi_i, \hat{\pi}_i]$. Assume $\pi_i$ is the linear mixture of $\pi_i, \hat{\pi}_i$, and $\bar{\pi}_i = \rho_i \pi_i + (1 - \rho_i)\hat{\pi}_i$, where $\rho_i \in [0, 1]$. Therefore, we can re-write the $g_i^{\pi_i, \pi_{-i}}(\pi_i, \pi_{-i})$ in the form of:

$$g_i^{\pi_i, \pi_{-i}}(\pi_i, \pi_{-i}) = g_i^{\pi_i, \pi_{-i}}(\rho_i, \rho_{-i}) = \rho_i(1 - \rho_{-i})g_i^{i, -\hat{i}} + (1 - \rho_i)\rho_{-i}g_i^{\hat{i}, -i} + (1 - \rho_i)(1 - \rho_{-i})g_i^{\hat{i}, -\hat{i}}. \tag{25}$$

Then we have:

$$\nabla_i g_i(\rho_{-i}) = (1 - \rho_{-i})g_i^{i, -\hat{i}} - \rho_{-i}g_i^{\hat{i}, -i} - (1 - \rho_{-i})g_i^{\hat{i}, -\hat{i}}, \tag{26}$$

and $\boldsymbol{\xi}(\pi_i, \pi_{-i}) = \boldsymbol{\xi}(\rho_i, \rho_{-i})$. Given a meta Nash policy pair $(\bar{\pi}_i, \bar{\pi}_{-i})$, where $\bar{\pi}_i = \bar{\rho}^i \pi_i + (1 - \bar{\rho}^i)\hat{\pi}_i$, according to the Nash definition, we have:

$$\begin{pmatrix} \bar{\rho}_i \\ 1 - \bar{\rho}_i \end{pmatrix}^T \begin{pmatrix} g_i^{i, -i} & g_i^{i, -\hat{i}} \\ g_i^{\hat{i}, -i} & g_i^{\hat{i}, -\hat{i}} \end{pmatrix} \begin{pmatrix} \bar{\rho}_{-i} \\ 1 - \bar{\rho}_{-i} \end{pmatrix} \geq \begin{pmatrix} \rho_i \\ 1 - \rho_i \end{pmatrix}^T \begin{pmatrix} g_i^{i, -i} & g_i^{i, -\hat{i}} \\ g_i^{\hat{i}, -i} & g_i^{\hat{i}, -\hat{i}} \end{pmatrix} \begin{pmatrix} \bar{\rho}_{-i} \\ 1 - \bar{\rho}_{-i} \end{pmatrix}, \tag{27}$$

which implies:

$$\begin{aligned} (\bar{\rho}_i - \rho_i)\nabla_i g_i(\bar{\rho}_{-i}) \geq 0, & \quad \bar{\rho}_i, \forall \rho_{-i} \in [0, 1], \\ (\bar{\rho}_{-i} - \rho_{-i})\nabla_{-i} g_{-i}(\bar{\rho}_i) \geq 0, & \quad \bar{\rho}_i, \forall \rho_{-i} \in [0, 1]. \end{aligned} \tag{28}$$

When $\bar{\rho}_i, \bar{\rho}_{-i} \in (0, 1)$ in accordance with the Nash condition in Eq. 28, $\nabla_i g_i(\bar{\rho}_{-i}) = \nabla_{-i} g_{-i}(\bar{\rho}_i) = 0$. It shows that $(\bar{\pi}_i, \bar{\pi}_{-i})$ is a fixed point due to $\boldsymbol{\xi}(\bar{\pi}_i, \bar{\pi}_{-i}) = \boldsymbol{\xi}(\bar{\rho}_i, \bar{\rho}_{-i}) = \mathbf{0}$. For the boundary case, where $\bar{\rho}_i$ or $\bar{\rho}_{-i} \in \{0, 1\}$, because they are constrained to the unit square $[0, 1] \times [0, 1]$, the gradients on the boundaries of the unit square are projected onto the unit square, which means additional points of zero gradient exist. In other words, $\nabla_i g_i$ and $\nabla_{-i} g_{-i}$ are still equal to zero in boundary case, and the $(\bar{\pi}_i, \bar{\pi}_{-i})$ is a fixed point in both cases.

Next, we determine what types of the fixed point that $(\bar{\pi}_i, \bar{\pi}_{-i})$ belongs to. According to the Eq. 24, we have the exact Hessian Matrix for the restricted game:

$$H = \nabla \xi = \begin{pmatrix} 0 & g_i^{\hat{i}, -\hat{i}} - g_i^{i, -\hat{i}} - g_i^{\hat{i}, -i} \\ g_{-i}^{\hat{i}, -\hat{i}} - g_{-i}^{i, -\hat{i}} - g_{-i}^{\hat{i}, -i} & 0 \end{pmatrix} \tag{29}$$

The eigenvalue $\lambda$ of $H$ can be computed:

$$\lambda^2 - \text{Tr}(H)\lambda + \det(H) = \lambda^2 - (g_i^{\hat{i}, -\hat{i}} - g_i^{i, -\hat{i}} - g_i^{\hat{i}, -i})(g_{-i}^{\hat{i}, -\hat{i}} - g_{-i}^{i, -\hat{i}} - g_{-i}^{\hat{i}, -i}) = 0 \tag{30}$$

Denotes $\bar{g}_i := g_i^{\hat{i}, -\hat{i}} - g_i^{i, -\hat{i}} - g_i^{\hat{i}, -i}$, we have $\boldsymbol{\lambda} = \pm\sqrt{\bar{g}_i \bar{g}_{-i}}$. Therefore, we can have following cases for the fixed point $(\bar{\rho}_i, \bar{\rho}_{-i})$:

1. Fully cooperative games: $\bar{g}_i \leq 0, \bar{g}_{-i} \leq 0$, then $H(\bar{\rho}_i, \bar{\rho}_{-i}) \preceq 0$, which means $(\bar{\rho}_i, \bar{\rho}_{-i})$ is a stable fixed point as we are maximizing the objective.

2. Fully competitive games: $\bar{g}_i > 0, \bar{g}_{-i} < 0$ or $\bar{g}_i < 0, \bar{g}_{-i} > 0$, all $\boldsymbol{\lambda}$ have two pure imaginary eigenvalues with zero real part, where $(\bar{\rho}_i, \bar{\rho}_{-i})$ is a saddle point.

3. General sum games: they are in-between the cooperative and competitive games, which means $(\bar{\rho}_i, \bar{\rho}_{-i})$ can be either stable fixed point or saddle point.

Because we assume $\hat{\pi}_i$ monotonically improved compared to $\pi_i$, then even in zero-sum case, there is at least one negative value in $\bar{g}_i$ and $\bar{g}_{-i}$. Therefore, in all the situations, $(\bar{\rho}_i, \bar{\rho}_{-i})$ is not unstable, and could be a stable point or saddle point. We define them as a weak stable fixed point. It also has a tighter lower bound than the independent trust region improvement seen in Remark 1:

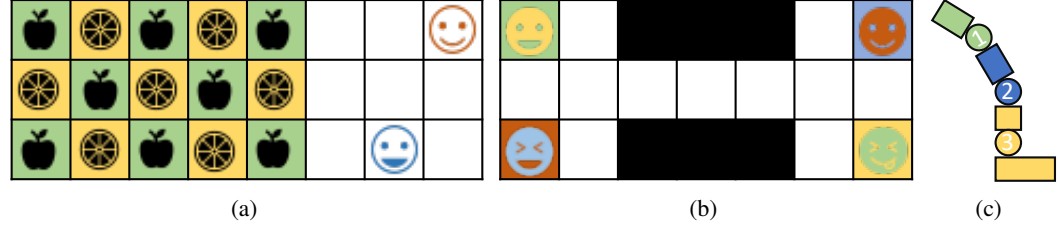

Figure 8: Multi-agent discrete and continuous action tasks: (a) 2-agent checker (discrete), (b) 4-agent switch (discrete), (c) 3-agent MuJoCo hopper (continious).

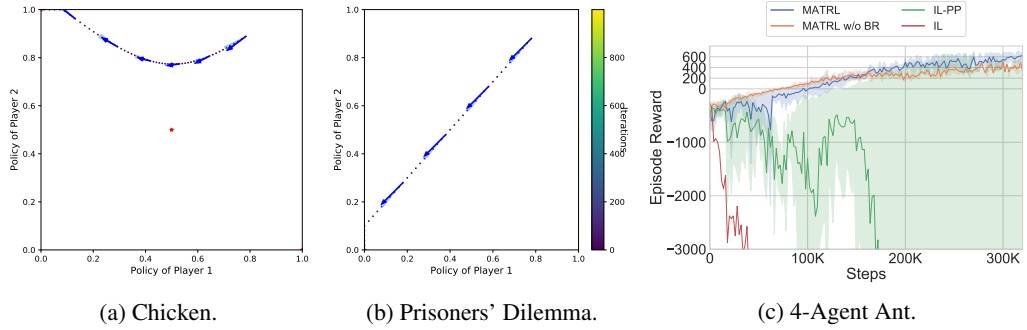

(a) Chicken.      (b) Prisoners' Dilemma.      (c) 4-Agent Ant.

Figure 9: **(a)-(b)**: Learning dynamics of two games using MATRL. **c**: Extra learning curves in 4-Agent Ant multi-agent MuJoCo task.

**Remark 1.** *Let $(\rho_i, \rho_{-i})$ be a Nash equilibrium of the policy-space meta-game $\mathcal{M}(\pi_i, \hat{\pi}_i, \pi_{-i}, \hat{\pi}_{-i})$, which is used for computing the linear mixture policies $\bar{\pi}_i, \bar{\pi}_{-i}$. For simplicity, define $\bar{\rho}_i = 1 - \rho_i$, then we have the payoff improvement lower bound for $\bar{\pi}_i, \bar{\pi}_{-i}$:*

$$\eta_i(\bar{\pi}_i, \bar{\pi}_{-i}) - \eta_i(\pi_i, \pi_{-i}) \geq g_i^{\pi_i, \pi_{-i}}(\bar{\pi}_i, \bar{\pi}_{-i}) - \frac{4\gamma\epsilon_i}{(1-\gamma)^2}(\alpha_i\bar{\rho}_i + \alpha_{-i}\bar{\rho}_{-i} - \alpha_i\alpha_{-i}\bar{\rho}_i\bar{\rho}_{-i})^2, \quad (31)$$

*that is a tighter lower bound compared with Theorem 1.*

Finally, we obtain MATRL as follows: First, an agent $i$ collects a set of trajectories using its current policy $\pi_i$ by independent play with other agents. Then a predicted policy $\hat{\pi}_i$ can be estimated using the single-agent trust region methods, which has a trust payoff improvement against the other agents' current policy $\pi_{-i}$. However, this trust payoff improvements would not benefit convergence requirements for the multi-agent system due to other agents adaptive learning. To solve this problem, we approximate a $n$-agent two-action meta-game in policy-space by reusing the trajectories from the last TPR step. In this game, each agent $i$ has two pure strategies: choosing the *current policy* $\pi_i$ or *predicted policy* $\hat{\pi}_i$ and the corresponding payoffs are the expected advantages (defined in Eq. 3) of the joint policy pairs. By constructing such a meta-game, we transform a complex multi-agent interactions problem into game-theoretic analysis concerning the underlying game restricted in $[\pi_i, \hat{\pi}_i]$. Then we can obtain a weak stable fixed point as TSR within the TPR by solving the meta-game,. When the fixed point is a saddle point we then take the best response to the weak stable fixed point to get the next iteration's policies. This encourages exploration and avoid stagnation at an unexpected saddle point.

# D ENVIRONMENT DETAILS

**Random 2 × 2 Matrix Games**. We created a generator of $2 \times 2$ matrix games based on the category provided by (Pangallo et al., 2017). Coordination games have characteristics enabling one agent to improve the payoff without decreasing the payoff of the other agent. Anti-coordination games are ones where one agent improves the payoff while the other agent's payoff decreases. Both coordination and anti-coordination games can have two pure NEs and one mixed strategy NE. In cyclic games, the action selections of agents that is based on their actions will form a cycle, ensuring that there is no pure NE in the game. Instead only mixed strategy NE will be found.

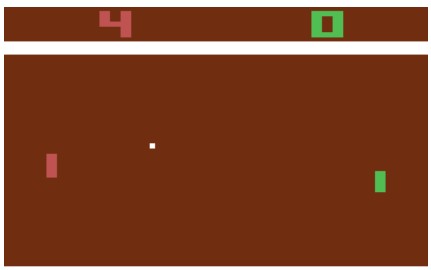

Figure 10: Pong game in Atari 2600.

**Grid World Games**. In two-player checker, as shown in Fig. 8a, there is one sensitive player who gets reward 5 when they collect an apple and 5 when they collect a lemon; a less sensitive player gets 1 for apple and 1 for lemon. The learning goal is to let the sensitive player get apples and the other one get lemons to have a higher total reward. In four-player switch, as shown in Fig. 8b, to reach the targets, agents need to figure out a way to go through a narrow corridor. The agent gets $-1$ for taking each step and 5 when arriving at a target. Four-player switch uses the same map as two-player switch, where two agents start from the left side and the others from the right side to go through the corridor to reach the targets. With more agents in four-player switch, learning becomes more challenging. MATRL agents achieved higher total rewards compared to baseline algorithms within the same number of steps.

**Multi-Agent MuJoCo Tasks**. We used the three-agent Hopper environment described in (de Witt et al., 2020), and Fig. 8c, where three agents control three joints of the robot and learn to cooperate to move forward as far as possible. The agent is rewarded by the number of time steps that they move without falling. Each agent has 3 continuous output values as the action, and all the agents have a full observation of the states of size 17. We use the same hyper-parameters for MATRL, MATRL w/o BR, and IL-PP. For MADDPG agent, we use the hyper-parameters described in the paper (de Witt et al., 2020).

**Multi-Agent Atari Game**. The pong game is a multi-agent Atari version[2] of table tennis, as shown in Fig. 10. Two players must prevent a ball from whizzing past their paddles and allowing their opponent to score. The game ends when one side earns 21 points.

## E    EXPERIMENTAL PARAMETER SETTINGS

For all the tasks, the most important hyper-parameters are learning rate/step size, the number of update steps, batch size and value, policy loss coefficient. Appropriate learning rate and update steps plus larger batch size give a more stable learning curve. And for different environments, policy and value network loss coefficients that keep two losses at the same scale are essential in improving the learning result and speed. Also, for meta-game construction and best response update where we use the importance ratio to do estimation, a clipping factor of the ration is vital to achieving a stable and monotonic improving result. The followings are the detailed parameter settings for each task.

**Matrix Game and Random** $2 \times 2$ **Matrix Games**. The hyper-parameters settings for MATRL, IGA-PP, and WoLF are listed in Table 2. As shown in Fig. 9, we also listed the additional convergence analysis in classical Chicken and Prisoners' Dilemma Games, which demonstrate good convergence performance of MATRL on both games. For MATRL, we have the KL-divergence coefficient as an extra hyper-parameter to add the KL-divergence as part of the loss in policy updating. And for the baseline algorithm WoLF, we give the real NE of the game as part of the parameters. In all the games, all the algorithms shared the same initial policy values $[0.9, 0.1]$ for player 1 and $[0.2, 0.8]$ for player 2.

**Grid World Games and Multi-agent Continuous Control Task**. The hyper-parameters settings for MATRL are given in Table 3. We used the same hyper-parameters for MATRL, MATRL w/o BR, IL-PP, and IL. The only difference is whether to use Best Response and the meta-game or not. We

---

[2]https://github.com/PettingZoo-Team/Multi-Agent-ALE

Table 2: Hyper-parameter settings in $2 \times 2$ matrix games.

| SETTINGS | VALUE | DESCRIPTION |
|---|---|---|
| **COMMON SETTINGS** | | |
| INITIAL POLICIES 1 | $[0.9, 0.1]$ | THE INITIAL POLICY VALUES FOR PLAYER 1 |
| INITIAL POLICIES 2 | $[0.2, 0.8]$ | THE INITIAL POLICY VALUES FOR PLAYER 2 |
| **MATRL SETTINGS** | | |
| BEST RESPONSE LEARNING RATE | 0.03 | THE LEARNING RATE FOR THE BEST RESPONSE STEP |
| KL COEFFICIENT | 100 | THE KL-DIVERGENCE COEFFICIENT IN POLICY LOSS |
| **WOLF SETTINGS** | | |
| LEARNING RATE MAXIMUM | 0.06 | THE MAXIMUM LEARNING RATE FOR WOLF LEARN FAST AGENT |
| LEARNING RATE MINIMUM | 0.02 | THE MINIMUM LEARNING RATE FOR WOLF WIN AGENT |

Table 3: MATRL hyper-parameter settings in grid worlds.

| COMMON SETTINGS | VALUE | DESCRIPTION |
|---|---|---|
| POLICY LEARNING RATE | 0.002 | OPTIMIZER LEARNING RATE. |
| BATCH SIZE | 2000 | NUMBER OF DATA POINT FOR EACH UPDATE. |
| GAMMA | 0.99 | LONG TERM DISCOUNT FACTOR. |
| HIDDEN DIMENSION | 128 | SIZE OF HIDDEN STATES. |
| NUMBER OF HIDDEN LAYERS | 2 | NUMBER OF HIDDEN LAYERS. |
| NASH EQUILIBRIUM SOLVER METHOD | CMAES | THE METHOD FOR FINDING THE NASH EQUILIBRIUM OF META-GAME |
| NEURAL NETWORK | MLP | THE NEURAL NETWORK ARCHITECTURE FOR POLICY AND CRITIC |
| POLICY UPDATE ITERATIONS | 10 | NUMBER OF GRADIENT STEPS FOR EACH BATCH OF UPDATE. |
| BEST RESPONSE LEARNING RATE | 0.002 | THE LEARNING RATE FOR BEST RESPONSE STEP |
| BEST RESPONSE INTERACTIONS | 5 | NUMBER OF GRADIENT STEPS FOR BEST RESPONSE STEP |
| KL COEFFICIENT | 0.001 | THE KL DIVERGENCE COEFFICIENT IN CALCULATING LOSS |
| ENTROPY COEFFICIENT | 0.05 | THE ENTROPY COEFFICIENT IN CALCULATING LOSS |
| POLICY RATIO CLIP | 0.1 | THE CLIP VALUE FOR POLICY RATIO |
| BEST RESPONSE IMPORTANCE RATIO CLIP | 0.1 | THE CLIP VALUE FOR BEST RESPONSE IMPORTANCE WEIGHT |
| **2 PLAYER SWITCH** | | |
| VALUE LOSS COEFFICIENT | 0.01 | THE VALUE LOSS IS LARGER THAN POLICY LOSS |
| **2 PLAYER CHECKER** | | |
| VALUE LOSS COEFFICIENT | 1.0 | THE VALUE LOSS IS AT SAME SCALE AS POLICY LOSS |
| **4 PLAYER SWITCH** | | |
| VALUE LOSS COEFFICIENT | 0.01 | THE VALUE LOSS IS LARGER THAN POLICY LOSS |

used Leaky ReLU as the activation function for both policies and value networks. For the training, we used paralleled workers to collect experience data and update the network weights separated then synchronize all the works to have the final updated weights. We used different value loss and policy loss coefficients to balance the weights of two losses. For the Switch games, we used small value loss coefficients because the value loss is between $[0 - 10]$ while the absolute value policy loss is smaller than $1e - 2$. For the Checker game, the value loss and policy loss are at the same scale between $[1e - 4, 1e - 2]$. Also, we added entropy loss and KL loss to encourage exploration and limit the policy update for each step. We used (Šebek, 2013) as the Nash equilibrium solver for finding the meta-game Nash. The Nash solver is CMAES for all the experiments. If not particularly indicated, all the baselines use common settings as listed in Table 3. VDN, QMIX use common individual action-value networks as those used by MATRL; each consists of two 128-width hidden layers. We includes more experiment result on 4-agent ant task multi-agent MuJoCo task in Fig. 9c, which also demonstrate the superior performance of MATRL compared to other settings. The specialized parameter settings for each algorithm are provided in Table 4 and 5:

**Multi-agent Atari Pong**. The hyper-parameters setting for MATRL are listed in Table 6. We used the same hyper-parameters for MATRL and IL. We take the raw pixel input from the Atari environment, and we processed it with a convolution network, which has filter sizes [8,4,3], kernel sizes (3,3,3), and stride sizes [4,2,1] and "VALID" as padding. Then we pass the processed embedding to a 2 layer fully connected network to get the policy.

Table 4: Hyper-parameter settings for baseline algorithms in grid worlds.

| SETTINGS | VALUE | DESCRIPTION |
|---|---|---|
| **VDN** | | |
| MONOTONE NETWORK LAYER | 2 | LAYER NUMBER OF MONOTONE NETWORK. |
| MONOTONE NETWORK SIZE | 128 | HIDDEN LAYER SIZE OF MONOTONE NETWORK. |
| TARGET NETWORK UPDATE INTERVAL | 200 | NUMBER OF ITERATIONS BETWEEN EACH TARGET NETWORK UPDATE |
| LEARNER | DOUBLE-Q LEARNER | THE ALGORITHMS FOR EACH AGENT |
| **QMIX** | | |
| JOINT ACTION-VALUE NETWORK LAYER | 2 | LAYER NUMBER OF JOINT ACTION-VALUE NETWORK. |
| JOINT ACTION-VALUE NETWORK SIZE | 128 | HIDDEN LAYER SIZE OF JOINT ACTION-VALUE NETWORK. |
| LEARNER | DOUBLE-Q LEARNER | THE ALGORITHMS FOR EACH AGENT |

Table 5: Hyper-parameter settings in multi-agent MuJoCo hopper.

| SETTINGS | VALUE | DESCRIPTION |
|---|---|---|
| **MATRL AND ITS VARIANTS** | | |
| AGENT ALGORITHM | PPO | THE LEARNING ALGORITHM FOR AGENT |
| NETWORK | 2 LAYER MLP [128, 128] | THE NETWORK ARCHITECTURE AND SIZE FOR THE PPO AGENT |
| LEARNING RATE | 0.002 | LEARNING RATE FOR AGENTS |
| BATCH IZE | 4000 | BATCH SIZE FOR ONE UPDATE |
| VALUE LOSS COEFFICIENT | 0.001 | THE VALUE LOSS COEFFICIENT IN TOTAL LOSS |
| POLICY LOSS COEFFICIENT | 100 | THE POLICY LOSS COEFFICIENT IN TOTAL LOSS |
| POLICY UPDATE ITERATIONS | 10 | NUMBER OF GRADIENT STEPS FOR EACH BATCH OF UPDATE. |
| BEST RESPONSE LEARNING RATE | 0.002 | THE LEARNING RATE FOR BEST RESPONSE STEP |
| BEST RESPONSE INTERACTIONS | 5 | NUMBER OF GRADIENT STEPS FOR BEST RESPONSE STEP |
| ENTROPY COEFFICIENT | 0.05 | THE ENTROPY COEFFICIENT IN TOTAL LOSS |
| KL-DIVERGENCE COEFFICIENT | 0.01 | THE KL-DIVERGENCE COEFFICIENT IN TOTAL LOSS |
| GAMMA | 0.99 | DISCOUNT FACTOR |
| **MADDPG** | | |
| NETWORK | 2 LAYER MLP [300, 300] | THE NETWORK ARCHITECTURE AND SIZE FOR THE PPO AGENT |
| LEARNING RATE | 0.001 | LEARNING RATE FOR AGENTS |
| BATCH SIZE | 100 | BATCH SIZE FOR ONE UPDATE |
| UPDATE INTERVAL | 100 | UPDATE THE NETWORK EVERY 100 TIME STEPS |
| PRE-TRAIN TIMETEPS | 10000 | NUMBER OF TIME STEPS BEFORE NETWORK UPDATE |
| GAMMA | 0.99 | DISCOUNT FACTOR |

Table 6: Hyper-parameter settings in multi-agent pong Atari.

| SETTINGS | VALUE | DESCRIPTION |
|---|---|---|
| **MATRL AND ITS VARIANTS** | | |
| AGENT ALGORITHM | PPO | THE LEARNING ALGORITHM FOR AGENT |
| NETWORK | 3 LAYER CNN, 2 LAYER FC | THE NETWORK ARCHITECTURE AND SIZE FOR THE PPO AGENT |
| LEARNING RATE | 0.002 | LEARNING RATE FOR AGENTS |
| BATCH IZE | 4000 | BATCH SIZE FOR ONE UPDATE |
| VALUE LOSS COEFFICIENT | 0.1 | THE VALUE LOSS COEFFICIENT IN TOTAL LOSS |
| POLICY LOSS COEFFICIENT | 10 | THE POLICY LOSS COEFFICIENT IN TOTAL LOSS |
| POLICY UPDATE ITERATIONS | 10 | NUMBER OF GRADIENT STEPS FOR EACH BATCH OF UPDATE. |
| BEST RESPONSE LEARNING RATE | 0.002 | THE LEARNING RATE FOR BEST RESPONSE STEP |
| BEST RESPONSE INTERACTIONS | 5 | NUMBER OF GRADIENT STEPS FOR BEST RESPONSE STEP |
| ENTROPY COEFFICIENT | 0.05 | THE ENTROPY COEFFICIENT IN TOTAL LOSS |
| KL-DIVERGENCE COEFFICIENT | 0.01 | THE KL-DIVERGENCE COEFFICIENT IN TOTAL LOSS |
| GAMMA | 0.99 | DISCOUNT FACTOR |

