# OpenReview forum: "Multi-Agent Trust Region Learning"
_ICLR.cc/2021/Conference — Reject_

### Official Review · AnonReviewer3 · 2020-10-17
**Interesting idea but has a number of holes**

**Rating:** 4
**Confidence:** 4

**Review:**

Summary:
This paper proposes a modification of the Independent Learners trust region policy optimization method in general sum games. The modification consists of first forming a “meta game”—ie. the matrix game in which each agent’s options are his previous policy and his independent trust region optimized policy—-and then interpolating between the two for each agent according to a Nash equilibrium of the meta game. The paper shows that this algorithm results in each step generating a “weak stable fixed point”. The paper concludes by showing a number of experimental results indicating the convergence and overall performance of the method as compared with relevant baselines in a number of different games.

Strengths:
* good motivation of why independent learner algorithms may not converge due to missed coupling
* clever construction of meta game
* good choice of variety of experiments

Weaknesses:
* the notion of “weak stable fixed point” is extremely weak. For example , Mazumdar 2020 (who is cited) and many others consider much stronger criteria like local Nash or differential Nash, or even quasi-Nash. As I understand it, “weak stability” is just requiring that iterates don’t ever find a local minimum in the coupled strategy space of all agents. Not only is this artificially introducing a coupling requirement between the agents, but it is basically just saying that the agents shouldn’t ever find themselves all wishing to change collectively.
* the preliminaries are sometimes stated in a way that indicates only two players and other times N players
* figures 1 and 2 are confusing
* I find theorem 1 to be hard to parse (much notation is not defined clearly), based on very strong assumptions (alpha coupling, discrete spaces), essentially pointing out a fact about prior work on independent methods (not the current proposed one), and overall unnecessary in the present paper. I would advocate removing it.
* it should be clarified that the rho are really distributions corresponding to mixed Nash strategies in the meta game, rather than deterministic strategies (which may not even exist)
* in the experiments, it is not statistically useful to show results for only 5 seeds. I understand that these things are expensive and I have this comment for the vast majority of work in RL I have ever read, but showing plots like figure 5 is misleading, esp. without an immediate caution in the caption and remain text
* I found figure 6 and the related text very confusing and not really useful in understanding what is going on
* only validating convergence in matrix games is very limited. I was missing a more complete discussion of convergence, if only experimental

Nitpicks:
* there were numerous typos and syntax/semantic errors throughout the paper. I would recommend employing the services of a copy editor in future

Overall:
I like the main concept of this paper, but I really can not recommend it for publication at this time, I have listed a number of directions in which I feel this paper could be improved in rebuttal or even in a later submission if it goes that route. My main concerns are: (1) the weakness of the theoretical property of “weak stability”, (2) lack of a more complete evaluation of convergence, (3) general confusion throughout reading the paper.

---

> ### Author Response · Authors · 2020-11-16
> **Response to Reviewer 3**
>
> We thank the reviewer for the detailed feedback.
>
> **Question**: the notion of “weak stable fixed point” is extremely weak. For example , Mazumdar 2020 (who is cited) and many others consider much stronger criteria like local Nash or differential Nash, or even quasi-Nash.
>
> **Response**: At first, it is worth pointing out that local Nash is not a stronger notion than stable points. In fact, it is the other way round. A local Nash is only a stable fixed point in zero-sum games (see Lemma 8 in [1]). In potential games, a local Nash can easily be unstable; think about the case of x^2+y^2+2xy, local nash of (0,0) is not a stable point. In general, stable points is a set of solution concepts for gradient-based methods that try to describe the cases when the gradient of policy change is zero, while local nash requires the best response from each agent’s perspective.
> Also, we agree that “weak stable fixed point” is a weak requirement, but it is reasonable to avoid strict maxima/minima. We make weak stable requirements for two reasons: 1. We want the method suits for general game settings, as shown in Eq. 16 and its following text, in different game settings: cooperative, competitive, and general-sum, the fixed-point found by the meta-game analysis can be either stable or saddle points. If we make some additional game class assumptions, we can easily obtain a stronger requirement. 2. The motivation of this MATRL is to find a low-cost way to search for a point better than an independent trust region learner. So, in the MATRL case, we only consider the stable points within each trust region gradient-descent steps. And in fact, one can only know that in a multi-agent setting because when each agent is updating using a trust region method in one gradient step, it is impossible (unless additional constraints to force the hessian definite in Eq. 16 to be PSD) to guarantee all of them will converge to an even fully stable point or local Nash. Therefore, finding other better stable points in a restricted step could be a good idea, it may suit for next step’s work, but it comes with a cost, requiring additional computation or assumptions which may break the most general settings.
>
> [1] Balduzzi D, Racaniere S, Martens J, et al. The mechanics of n-player differentiable games[J].ICML, 2018.
>
>
> **Question**: a more complete discussion of convergence,
>
> **Response**: Thanks for the kind suggestions, it is a good point to clarify the method’s contribution and effectiveness.   We have added more global convergence discussion in section 3 and the convergence analysis in experiments. In short, from the global convergence side, MATRL is a variant of LookAhead methods (e.g., extra-gradient, LOLA, SOS) via best response to the trust stable region (one-step look-ahead), which enjoys local convergence and avoids strict saddles in all differentiable games, see [2].
>
> [2] Letcher A, Foerster J, Balduzzi D, et al. Stable opponent shaping in differentiable games. ICLR, 2019.
>
>
> Responses to other questions:
>
> According to your comments, we have refined the revision paper to make the it is easier to follow:
>
> 1. Changed figure 1 and added additional explanations in captions to understand figures 1 and 2 better.
>
> 2. For Figures 4, 5, 6, and table 1, added additional experiment settings and descriptions of results to make it easier to understand the results.
>
> 3. We have simplified theorem 1 and had an additional discussion to show that multi-agent independent improvement with worse lower bound than the single-agent case.
>
> 4. Added the mixed nash descriptions for $rho$.
>
> Working on changes:
>
> 1. Adding more runs for experiments that need much more time. We plan to increase the number of random seeds for each model to 10 to obtain more confident results.
>
> 2. Thank you for pointing out these typos. To keep the minimum changes in the revision, we did not do the deep editing on the current revision.  But we definitely will look for a professional copy editor to fix all typos and syntax/semantic errors and make the presentation more fluent in the final version.

---

> > ### Comment · AnonReviewer3 · 2020-11-18
> > **Response to Authors**
> >
> > Thank you for your response. A couple points:
> >
> > 1. I still am not sure I understand how weak stable points can be stronger than local Nash (seems like local Nash is saying "no player wants to change within a neighborhood" and weak stable is saying "no player sees a direction in which his cost changes at all", where local Nash includes cases where there are ascent (but not descent) directions of a player's cost). In any case, I follow your justification for why it is more challenging to use stronger conditions in general; I hope that this can be made more clear in the manuscript.
> > 2. The discussion of convergence is a very good addition. As is the mention of mixed Nash.
> > 3. 10 is certainly better than 5, but I would still suggest adding more clear warnings against naively drawing too much statistical power from these (or any similarly limited) results.
> >
> > My estimation of this paper is certainly improved, and will be reflected in discussions with other reviewers and the editor.

---

> > > ### Author Response · Authors · 2020-11-20
> > > **Further Response to Reviewer 3**
> > >
> > > Thanks for your further feedback!
> > >
> > > **Question**: I still am not sure I understand how weak stable points can be stronger than local Nash.
> > >
> > > **Response**: Sorry for the confusion still. We didn’t mean that weak stable is stronger than local Nash; we think the stable fixed points and local Nash are not equivalent concepts in all the cases. In different cases (like Fig. 1 in [1]), a local Nash can be either a stable fixed point or saddle point. Similarly, a weak stable fixed point can be a stable or saddle point as well. Therefore, we think which concept would be better will depend on the specific objectives or settings, and we have updated the revision accordingly to justify this point.
> > >
> > > [1] Mazumdar E, Ratliff L J, Sastry S S. On Gradient-Based Learning in Continuous Games[J]. SIAM Journal on Mathematics of Data Science, 2020, 2(1): 103-131.

---

> > > ### Author Response · Authors · 2020-11-24
> > > **10 random seeds for the experiments.**
> > >
> > > Thanks for your kind suggestion.
> > >
> > > We have added 5 more random seeds for all the models in the experiments, and new learning curves with 10 seeds are updated in Fig. 5, the latest revision paper.

---

### Official Review · AnonReviewer4 · 2020-10-27
**Novel algorithm addressing an important problem.**

**Rating:** 8
**Confidence:** 4

**Review:**

---- Summary ----
This paper proposes a multi-agent learning algorithm where after each policy improvement step, a small meta-game is analysed to propose a corrected step. The paper performs experiments in various MARL environments to test the approach.

---- Reasons for score ----
I recommend accepting this paper. The paper addresses the important problem of the non-convergence of independent learning in MARL. The algorithm proposed is novel and well justified, and the experiments show that it yields an improvement over independent learners.

---- Pros ----
1. The fundamental idea of the paper - taking independent policy improvement steps, and then analysing a local metagame to decide how to update policies - is interesting and novel.
2. The practical algorithm this leads to helps with the problems of independent learning in MARL, without a large increase in algorithm complexity.
3. The experiments demonstrate the advantages of the algorithm, in a variety of domains and with a fair comparison to the IL methods MATRL is being based on and other relevant baselines.
4. The paper is situated well with respect to the existing literature.

---- Cons ----
It seems to me that the effectiveness of the proposed method is likely to be heavily influenced by the underlying IL algorithm employed. In particular, an IL algorithm making very large policy updates is likely to need more correcting. For this reason, it would be interesting to investigate how the performance of MATRL and IL varies with a range of policy update sizes, and it would also be useful to clarify how the hyperparameters for the various experiments were selected.

---- Typos and other minor comments ----
1. Please clarify the meaning of the error bars in Figure 5.
2. In the sentence before section 3.1, steps->step and details->detail.
3. I didn't understand this in section 3.1: "We set agent i’s to make a monotonic improvement of its policy."
4. Near the bottom of page 3, "conflict interests" -> "conflicting interests"

---

> ### Author Response · Authors · 2020-11-16
> **Response to Reviewer 4**
>
> We thank the reviewer for the detailed feedback.
>
> **Question**: it would be interesting to investigate how the performance of MATRL and IL varies with a range of policy update sizes, and it would also be useful to clarify how the hyperparameters for the various experiments were selected.
>
> **Response**: It is a good suggestion to reveal further how and when MATRL works. As we assume the predicted policies have monotonic improvements, it is better to use trust-region-based ILs and small step size to make the meta-game easily estimated and solved.
> As for the hyperparameters selection, the most important hyper-parameters are learning rate/step size, the number of update steps, batch size and value, policy loss coefficient. Appropriate learning rate and update steps plus larger batch size give a more stable learning curve. And for different environments, policy and value network loss coefficients that keep two losses at the same scale are essential in improving the learning result and speed. Also, for meta-game construction and best response update where we use the importance ratio to do estimation, a clipping factor of the ratio is vital to achieving a stable and monotonic improving result. The followings are the detailed parameter settings for each task. We have included this discussion in Appendix E.

---

### Official Review · AnonReviewer1 · 2020-10-27
**Blind review**

**Rating:** 5
**Confidence:** 4

**Review:**

This paper presents a new trust-region method for multi-agent reinforcement learning (MARL). This approach extends ideas from single-agent trust-region methods to construct a smaller meta-game representing possible policy changes for each agent. The meta-game can then be solve to provide policy updates for the agents. Theory is provided for the meta-game (and corresponding restricted underlying game) and experimental results are shown with a number of baselines.

The idea of extending trust region methods to the multi-agent case is an interesting idea. As the paper points out, the multi-agent case is more complicated since the other agents are also learning, making the environment appear non-stationary from a single agent's perspective. The idea of using a meta-game with limited (i.e., 2) policy update choices at each step greatly simplifies learning, while at least somewhat taking other agent changes into account. Also, the method is based on theory for converging to a (weak stable) fixed point.

There are a lot of interesting ideas in the paper and the experiments are promising, but many of the details in the paper are not clear.

For example, the general assumptions and algorithm become clear in Figure 3 and Algorithm 1, but these should be explicitly stated much earlier (e.g., at the beginning of 3). Even so, it still isn't clear exactly what is assumed to be known about other agent policies. It appears that the process is fully centralized, where all agents know all other agents policies at all times. If this is indeed the case, why is it reasonable to use a centralized approach in a self-interested setting? In MARL, even assuming other agent actions are observed is a strong assumption, so assuming other agents policies are known is very strong. Furthermore, the paper says "MATRL also provides fully decentralized execution and only requires a centralized mechanism to adjust the step size rather than a centralized critic or communication." but it appears that the training phase would need much more than that (to solve the meta-game) while the execution could be decentralized. A more detailed description of the algorithm should be given along with the assumptions needed.

It also isn't clear how useful the theory is since it only applies to the restricted underlying game rather than the full game. Since it appears to assume the other agents are fixed, it would be useful to at least speculate what could be said about the full case (e.g., when other agents are not fixed and when it can't be assumed the other other agent policy updates are known). Also, the game values in 5 would just be estimates in the MARL case (due to sampling and other agent changes) so it doesn't seem like the theory would apply in that case. These issues should be clarified in the paper.

The experimental results are impressive and show the proposed method works well. The method (MATRL) converges faster or to better values in all the domains. Nevertheless, it is not clear how the domains were chosen and many of the baselines are weak. In terms of the domains, the paper should discuss why a subset of domains is chosen (i.e., why these particular domains). In terms of baselines, there are stronger methods in each case. For example, many other MARL methods would apply in the Checker, Switch and MuJoCo domains (e.g., COMA and MAAC). Why use only VND and QMIX (which are very different than the proposed method)?  Why not use any other method (besides independent learners) in multi-agent pong? Lastly, several of the domains are partially observable, but the method is only defined in terms of full observability and nothing is said about how it was adapted to run in the partially observable case. It is assumed that they pretend the observation is state, but this should be made clear.

---

> ### Author Response · Authors · 2020-11-16
> **Response to Reviewer 1**
>
> We thank the reviewer for the detailed feedback.
>
> **Question**:  It still isn't clear exactly what is assumed to be known about other agent policies.
>
> **Response**:  We are sorry for the confusion. To clarify the model assumptions,  we have made the “to estimate and solve the meta-game, the knowledge about other agents’ policies to collect the joint experience/trajectories is required” assumption more obvious at the beginning of section 3 (and section 3.2) and added more detailed comparisons with other CTDE methods in related work.
>
> **Question**: It also isn't clear how useful the theory is since it only applies to the restricted underlying game rather than the full game.
>
> **Response**:  From the global convergence side, MATRL is a variant of LookAhead methods (e.g., extra-gradient, LOLA, SOS) via best response to the trust stable region (one-step look-ahead), which enjoys local convergence and avoids strict saddles in all differentiable games, see [1].
> On the other side, in a local restricted underlying game (during each meta-game analysis step), the meta-game equilibrium is an approximated equilibrium of the true underlying game, see [2].
>
> [1] Letcher A, Foerster J, Balduzzi D, et al. Stable opponent shaping in differentiable games. ICLR, 2019.
> [2] Tuyls K, Perolat J, Lanctot M, et al. Bounds and dynamics for empirical game theoretic analysis[J]. Autonomous Agents and Multi-Agent Systems, 2020, 34(1): 7.
>
> **Question**: The paper should discuss why a subset of domains is chosen (i.e., why these particular domains).
>
> **Response**: We chose these environments (random matrix games, grid-worlds, multi-agent mujuco and atari pong) to test the method in various general settings, including cooperative/competitive and continuous/discrete games. VDN/QMIX etc is for discrete cooperative games, MADDPG is for continuous action games, and PPO based IL works for all the cases. We have added a clarification in the experiment sections.
>
> **Question**: Why use only VND and QMIX (which are very different than the proposed method)?
>
> **Response**: We used VND and QMIX as baselines on discrete grid-worlds because they empirically work well compared to COMA and MAAC. To do more comprehensive comparisons, we have included QTRAN as a baseline in grid-worlds, and we are also working on having more runs to make the results more confident.
>
> **Question**: Why not use any other method (besides independent learners) in multi-agent pong?
>
> **Response**: We used PPO based ILs in the Pong game (discrete action, competitive game, so we cannot apply VND/QMIX/MADDPG); because our MATRL can have the same policy models and parameters as PPO based ILs, then we can examine the effectiveness of the designed centralized trust region learning procedures. We agree that including more baselines to have comparisons would be much better, and we are working on adding more baselines like independent DQN, MAAC. But this game takes more time to train; we will update the results once it has been done.
>
> **Question**: how it was adapted to run in the partially observable case.
>
> **Response**: Thanks for indicating that! You are right, the grid-worlds are partially observable, and it is better to have an additional clarification to make it clearer. We now have added a description of the gap between the grid-worlds and formulations.

---

> > ### Comment · AnonReviewer1 · 2020-11-19
> > **Response**
> >
> > Thanks for the response. The answers are helpful, but some things are still unclear.
> >
> > First, thanks for clarifying that other agent policies are assumed to be known. As you say, this is common in CTDE methods, but those typically focus on the cooperative case when it is reasonable to assume all agents would share information with each other. Why is this a reasonable assumption in the more general (possibly competitive) case?
> >
> > Second, I'm still not convinced about the global convergence theory. In particular, there is a connection between solving the meta-game that will limit possible solutions. Citation [2] is helpful here, but the details should be more clear and self-contained in the paper. Furthermore, the theory seems quite weak, so discussing how useful it is would be beneficial.
> >
> > Third, the experiments are still somewhat weak. The responses are helpful for understanding why certain methods and domains were used, but the paper should still contain more common domains and comparisons with more state-of-the-art methods.

---

> > > ### Author Response · Authors · 2020-11-20
> > > **Further Response to Reviewer 1**
> > >
> > > Thanks for your further feedback!
> > >
> > > **Question**: Why is this a reasonable assumption in the more general (possibly competitive) case?
> > >
> > > **Response**: We agree that it is unrealistic to assume knowledge about other agents' policies in many real-world competitive scenarios. But in some cases, it might only require decentralized execution; like generative adversarial networks (GAN), it is ok to expose the generator and discriminator's parameters/policies. Furthermore, we want to show that the proposed method can suit the general settings. For example, even in zero-sum games, it can still stabilize the simultaneous learning processes, like some other research (e.g., SGA [1], MADDPG[2]).
> > >
> > > [1] Balduzzi D, Racaniere S, Martens J, et al. The mechanics of n-player differentiable games[J]. ICML, 2018.
> > >
> > > [2] Lowe R, Wu Y I, Tamar A, et al. Multi-agent actor-critic for mixed cooperative-competitive environments[J]. Advances in neural information processing systems, 2017, 30: 6379-6390.
> > >
> > > **Question**: I'm still not convinced about the global convergence theory. In particular, there is a connection between solving the meta-game that will limit possible solutions. Citation [2] is helpful here, but the details should be more clear and self-contained in the paper. Furthermore, the theory seems quite weak, so discussing how useful it is would be beneficial.
> > >
> > > **Response**: The meta-game analysis enhances the classic LookAhead method with variable step size scaling (e.g., WoLF-IGA [3], GA-SPP[4]) or two time-scale update rule [5]. In our case, the step size is varying at each TPR, which is controlled by the restricted meta-game Nash. We have added this clarification in the new revision and will try to have a formal mathematical and more precise discussion in the final version.
> > >
> > > [3] Bowling M, Veloso M. Multiagent learning using a variable learning rate[J]. Artificial Intelligence, 2002, 136(2): 215-250.
> > >
> > > [4] Song X, Wang T, Zhang C. Convergence of multi-agent learning with a finite step size in general-sum games[J]. AAMAS, 2019.
> > >
> > > [5] Heusel M, Ramsauer H, Unterthiner T, et al. Gans trained by a two time-scale update rule converge to a local nash equilibrium[C]//Advances in neural information processing systems. 2017: 6626-6637.
> > >
> > > **Question**:  The paper should still contain more common domains and comparisons with more state-of-the-art methods.
> > >
> > > **Response**: We understand that the more different/complex domains and baselines make the results more convincing, and we are working on more affordable experiments like multi-agent atari games, poker, etc.

---

### Official Review · AnonReviewer2 · 2020-10-28
**Clearly written, sufficient number of experiments, but some possible points of improvement**

**Rating:** 6
**Confidence:** 3

**Review:**

MULTI-AGENT TRUST REGION LEARNING

Summary

The authors establish a trust-region method for multi-agent RL. The idea is to apply TRPO to each agent separately, but with a trust region whose properties for agent i, depend on those of agent not-i. The authors propose, at iteration k, to find a weak stable fixed point for a game M_k, defined by a matrix of payoffs given by the (expectation value of) advantages. They show that solving for the equilibrium of M_k yields policies that monotonically improve performance, i.e., each agent's payoff at least does not decrease.

The authors addresss the extra cost introduced by having to solve a game at each iteration, which does not scale well as the number of agents N grows.

They compare with independent learners (IL), and evaluate 3 synthetic experiments, multi-agent variations of Mujoco sims, and Atari pong. For games have fully cooperative rewards, but feature adversarial dynamics/rewards, the authors compare convergence rates (faster convergence in synthetic games), pair-wise win-rate (MATRL beats IL agents on pong), and overall reward-growth (MATRL improves reward faster in cooperative games).

Strengths

- The paper is written clearly, and the technique is relatively straightforward.
- The empirical evaluation seems sufficient in terms of showing MATRL improves over IL.

Weaknesses

- I would have liked to also see some comparison with techniques like LOLA, symplectic gradient adjustments, etc, as it's unclear what equilibrium MATRL converges to. Alternative (higher-order) methods don't necessarily converge to the equilibria of the original game -- does MATRL do so? Are MATRL solutions still a 'true' Nash equilibrium of the underlying game?
- I would have liked to see more reporting on the runtime of solving the meta-game. As the authors mention, this is a crucial issue in scaling this up, and it would be good to see some quantitative benchmark of run-time vs number of agents.

Questions

- Eq 2 has a typo? D(pi_i, pi_i)

---

> ### Author Response · Authors · 2020-11-16
> **Response to Reviewer 2**
>
> We thank the reviewer for the detailed feedback.
>
> **Question**: some comparison with techniques like LOLA, symplectic gradient adjustments, etc
>
> **Response**: We have MATRL-PP as a substitute of LOLA (uses Taylor expansion to t approximate the best response to the predicted policies), in which MATRL-PP takes true best response gradient steps to the predicted. We have made this more obvious in the experiment settings.
>
> **Question**: Alternative (higher-order) methods don't necessarily converge to the equilibria of the original game -- does MATRL do so? Are MATRL solutions still a 'true' Nash equilibrium of the underlying game?
>
> **Response**: From the global convergence side, MATRL is a variant of LookAhead methods (e.g., extra-gradient, LOLA, SOS) via best response to the trust stable region (one-step look-ahead), which enjoys local convergence and avoids strict saddles in all differentiable games, see [1].
> On the other side, in a local restricted underlying game (during each meta-game analysis step), the meta-game equilibrium is an approximated equilibrium of the true underlying game, see [2].
>
> [1] Letcher A, Foerster J, Balduzzi D, et al. Stable opponent shaping in differentiable games. ICLR, 2019.
> [2] Tuyls K, Perolat J, Lanctot M, et al. Bounds and dynamics for empirical game-theoretic analysis[J]. Autonomous Agents and Multi-Agent Systems, 2020, 34(1): 7.
>
> **Question**: I would have liked to see more reporting on the runtime of solving the meta-game.
>
> **Response**: Thanks for this suggestion. In the revision, we analyzed the cost of time of MATRL, MATRL without Best Response (MATRL w/o BR), Independent Learner with Policy Prediction (IL-PP) and Independent learner (IL) on 2 agents Checker, 3 agents Hopper, and 4 agents Switch game with 20K sampling steps. For all methods, they run the same number of 20,000 env steps and 50 gradient steps. As shown in Fig. 7, revision paper, we can see that the time consumption of each variation has no significant difference, MATRL takes around  10%-20% more time than the IL agents.
>
> **Question**: Eq 2 has a typo? D(pi_i, pi_i)
>
> **Response**: Thanks for pointing out this typo. It should be  $D(\pi_i, \hat{\pi}_i)$, and we have corrected it in the updated version.

---

### Author Response · Authors · 2020-11-16
**General Response to Reviewers' Comments**

We thank all the reviewers, ACs, and PCs for your efforts and constructive feedback.

As suggested, we have refined the revision paper, including a discussion about the convergence of global underlying game, model assumptions, the reasonability of the weak stable fixed point in the local restricted underlying game, etc. Besides, we have added an extra baseline for the experiments and an empirical comparison of the extra cost from the proposed method compared to independent learners with various agent numbers.

We are currently running several additional experiments and more random seeds as suggested, but it takes more time. We will post an update as soon as new results are available.

---

> ### Author Response · Authors · 2020-11-24
> **More random seeds for the experiments.**
>
> Dear reviewers, as promised, the experiments with five more random seeds have been updated in Fig. 5, the latest revision paper.

---

### Decision · Program_Chairs · 2021-01-07
**Final Decision**

**Decision:**

Reject

**Comment:**

There was some slight disagreement on the paper, but the majority of reviewers agree that although some answers of the authors on questions brought good clarification, other issues still remain problematic. Some of the assumptions remain unclear (w.r.t CDTE), and reviewers still have doubts about the global convergence and weak stable fixed point concept, that lack clear math details.
The experiments are also still a bit too immature, more comparison is needed, as well as an evaluation on other domains.